# Investigation of artificial cells containing the Par system for bacterial plasmid segregation and inheritance mimicry

Jingjing Zhao[1] & Xiaojun Han [1]

A crucial step in life processes is the transfer of accurate and correct genetic material to offspring. During the construction of autonomous artificial cells, a very important step is the inheritance of genetic information in divided artificial cells. The ParMRC system, as one of the most representative systems for DNA segregation in bacteria, can be purified and reconstituted into GUVs to form artificial cells. In this study, we demonstrate that the eGFP gene is segregated into two poles by a ParM filament with ParR as the intermediate linker to bind ParM and *parC*-eGFP DNA in artificial cells. After the ParM filament splits, the cells are externally induced to divide into two daughter cells that contain *parC*-eGFP DNA by osmotic pressure and laser irradiation. Using a PURE system, we translate eGFP DNA into enhanced green fluorescent proteins in daughter cells, and bacterial plasmid segregation and inheritance are successfully mimicked in artificial cells. Our results could lead to the construction of more sophisticated artificial cells that can reproduce with genetic information.

Cell division is a key feature of life[1], and it is crucial that daughter cells inherit accurate and correct genomic material to maintain the species[2]. Eukaryotes possess complicated chromosome segregation machinery in mitotic spindles[3–5]. In prokaryotes, relatively simple mechanisms are used to partition newly replicated DNA[6–9]. The ParMRC partitioning system is among the most representative systems for bacterial DNA segregation[6,10–12]. This system consists of three elements, namely, ParM (an actin-like protein), ParR (an adapter protein), and *parC* (a centromere-like site)[13–15]. ParM is an ATPase that can bind ParR, while ParR can specifically bind to *parC*. The genomic materials (plasmids) are segregated into opposite poles of cells by the polymerization of ParM with the help of ParR and *parC* in the presence of ATP[16]. This process is important because it ensures that genetic material is evenly distributed among daughter cells prior to division. In a previous study, bipolar elongation of ParM filaments was observed in a bacterial cell through immunofluorescence microscopy[17]. Through time-lapse fluorescence microscopy, researchers observed dynamic segregation of plasmids towards the poles of cells by ParM polymerization, which was facilitated by ParR[18].

Once plasmids reach the poles of the cell, the ParM filaments disassemble spontaneously and distribute plasmids to each daughter cell after cell division. In addition to the in vivo study on the function of the ParMRC system, the *parC*-beads were shown to move in the opposite direction following ParM polymerization with the help of ParR in vitro[19]. The ParMRC partitioning system was successfully reconstituted into water-in-oil droplets to realize in vitro DNA segregation[20], which provided the possibility for plasmid segregation and inheritance in artificial cells.

Building artificial cells with true-to-life functionality is an ambitious goal in synthetic biology[21–25]. The inheritance of genetic material into daughter cells after division is an essential step toward the construction of a minimal cell. The division of artificial cells is triggered by mechanical force[26], osmotic pressure[27,28], pH[29,30], and active molecules[31]. The vesicles are symmetrically split by the sharp edge of a wedge-shaped splitter inside a microfluidic channel, which results from mechanical force[26]. This method provided a strategy for symmetric, quick, efficient vesicle division. Phase separation of lipids in the membrane of artificial cells provides a division plane for dividing

[1]State Key Laboratory of Urban Water Resource and Environment, School of Chemistry and Chemical Engineering, Harbin Institute of Technology, Harbin, China. e-mail: hanxiaojun@hit.edu.cn

mother vesicles into two daughter vesicles through osmotic pressure in a controllable manner[28]. Oleic acid in the inner leaflet of the GUVs (POPC/oleic acid) bilayer was deprotonated by the enzymatic reaction of urea-urease inside the vesicle, which enabled inner leaflet area variation and consequently led to division with the help of osmotic pressure[30]. All the abovementioned vesicle divisions did not involve the natural division protein machinery. Bacterial division proteins, including MinC, MinD, MinE, FtsA, and FtsZ were reconstituted into lipid vesicles[32–34], attempting to divide artificial cells. Currently, the division of artificial cells is mostly designed for shape splitting, and some divisions are accompanied by DNA redistribution[27,31,35,36]. GUVs that contain DNA grow and divide into daughter vesicles by synthesizing and incorporating phospholipids in situ upon the addition of vesicular membrane precursors[31]. DNA inside the nucleus of eukaryote-like artificial cells is redistributed into two daughter artificial cells after division via osmotic stress[27]. To date, researchers have found that all DNA redistribution in daughter artificial cells is random and passive and does not involve protein machinery or further translation of genetic information into DNA.

In this work, we mimic bacterial plasmid segregation and inheritance by using the ParMRC system to segregate DNA at the two poles of artificial cells. Then, a division is initiated to generate daughter cells that inherit maternal genetic material upon light irradiation, and the process is aided by Ce6 molecules and changes in osmotic pressure. Enhanced green fluorescent proteins are expressed inside daughter cells in the presence of the PURE system. This paper opens an avenue

for researchers to mimic prokaryotic division with functional successive generations.

## Results

Figure 1 illustrates the mimicry of bacterial plasmid segregation and genetic information inheritance using an artificial cell that contains the ParMRC system and PURE system. These artificial cells (Fig. 1a) were obtained using the emulsion-transfer method. Upon laser irradiation (405 nm), ATP molecules enter GUVs through transient pores due to the presence of Ce6[35] (Supplementary Fig. 1) to trigger the polymerization of ParM for *parC*-eGFP DNA segregation (Fig. 1b). Under hypertonic conditions, GUVs are deformed into a dumbbell shape with *parC*-eGFP DNA at two poles (Fig. 1c). The ParM filaments are split by a laser (561 nm) at the center region of the dumbbell GUV (Fig. 1c); moreover, the GUV divides into two daughter GUVs that contain *parC*-eGFP DNA and the PURE system (Fig. 1d) with the assistance of Ce6 molecules. eGFP (enhanced green fluorescent protein) is expressed in the daughter GUV upon translation of the eGFP gene using a PURE (protein synthesis using recombinant elements) system at 37 °C (Fig. 1e). The model genetic information of eGFP is inherited by daughter cells via the ParMRC system. The experimental data are discussed in the following results section.

### Purification of the ParMRC system and its interactions

ParM and ParR were overexpressed in *Escherichia coli* BL21 (DE3) cells and purified using previously described protocols[37,38]. The molecular

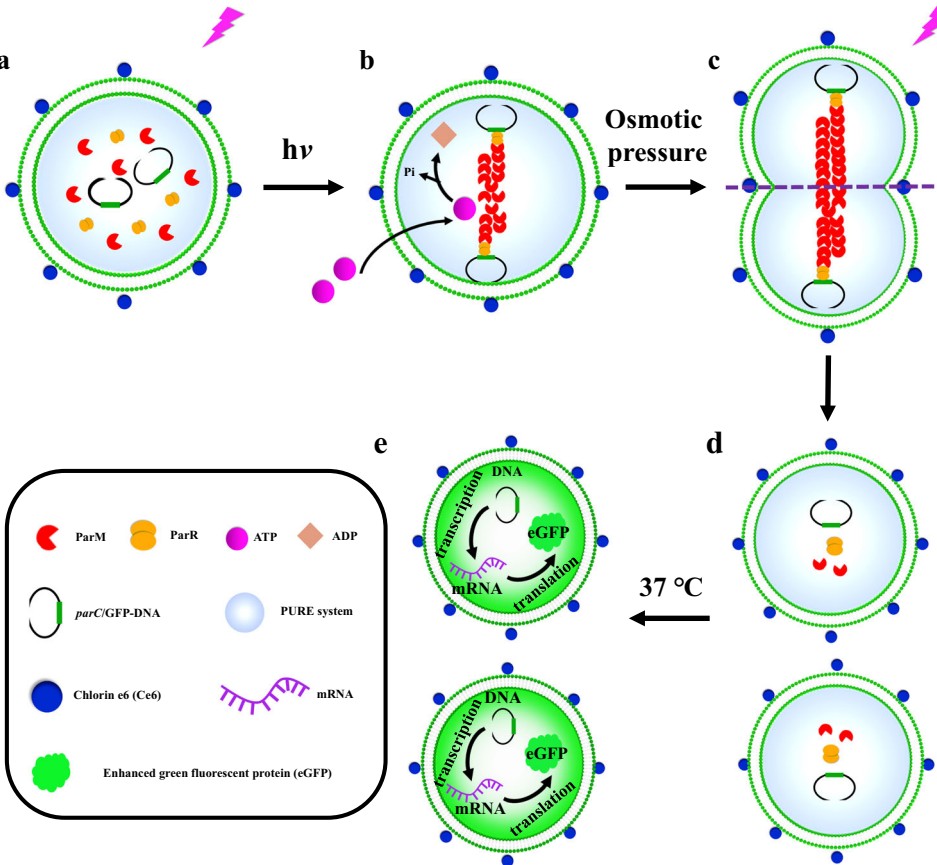

**Fig. 1 | Schematic illustration of artificial cells containing the ParMRC system and protein synthesis using recombinant elements (PURE) system for bacterial plasmid segregation and inheritance mimicry. a** Giant unilamellar vesicles (GUV) containing the ParMRC system and PURE system. **b** *parC*-eGFP DNA segregation by the polymerization of ParM triggered by adenosine triphosphate (ATP) influx upon laser irradiation. **c** A deformed GUV under hypertonic conditions. **d** Two daughter GUVs containing *parC*-eGFP DNA and the PURE system upon laser irradiation at the

center region of the deformed GUV (purple area in Fig. 1c). **e** Enhanced green fluorescent protein (eGFP) was expressed inside two daughter GUVs through translating the eGFP gene using a PURE (protein synthesis using recombinant elements) system at 37 °C. The PURE system contains ribosomes, amino acids, nucleoside triphosphates (NTPs), transfer ribonucleic acid (tRNAs), enzyme substrates, RNA polymerase, translation factors, and other necessary components.

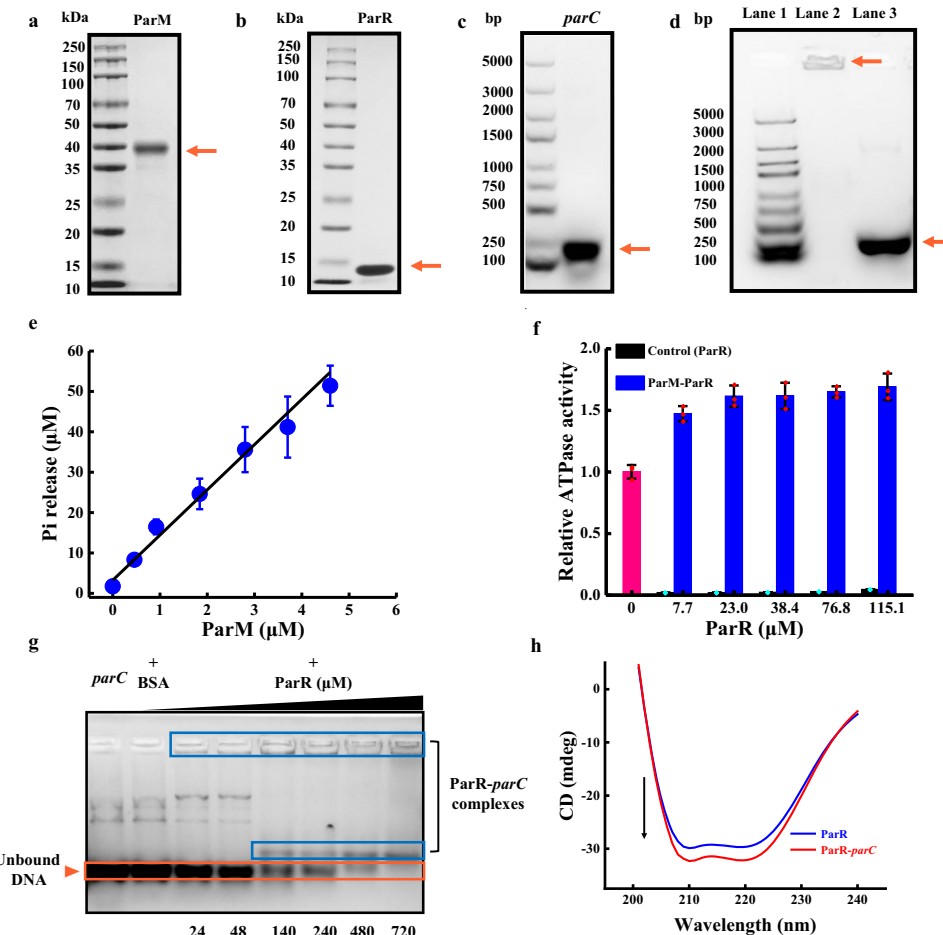

**Fig. 2 | Purification of the ParMRC system and its interaction.** Sodium dodecyl sulfate (SDS)-polyacrylamide gel images of ParM (39 kDa) (**a**) and ParR (14 kDa) (**b**). The bands depicted with orange arrows correspond to ParM and ParR, which have the expected molecular weights. $n = 3$ independent replicates. **c** Agarose gel image of biotinylated *parC*. The orange arrow indicates the band corresponding to *parC* (230 bp). $n = 3$ independent replicates. **d** Agarose gel image of the deoxyribonucleic acid (DNA) marker (Lane 1), biotinylated *parC*-beads (Lane 2), and *parC* (Lane 3). $n = 3$ independent replicates. **e** Phosphate release of ATP (4 mM) catalyzed by different ParM concentrations (0, 0.46, 0.92, 1.84, 2.8, 3.7, 4.6 μM) within 1 h. The phosphate release was obtained from three independent samples. The data were presented as the mean values ± SDs; $n = 3$ independent replicates. **f** The ATPase activity of ParM (4.6 μM) enhanced by ParR (7.7, 23.0, 38.4, 76.8, 115.1 μM); the pink, black, and blue columns represent the ATPase activity of ParM, ParR, and ParM-ParR, respectively. The ATPase activities of ParM, ParR, and ParM-ParR were obtained from three independent samples. The data were presented as the mean values ± SDs; $n = 3$ independent replicates. **g** Electrophoretic mobility shift assay (EMSA) of *parC* (275 ng) with increasing concentrations of ParR (24, 48, 140, 240, 480, and 720 μM). $n = 3$ independent replicates. **h** Far-UV circular dichroism spectra of ParR (3.31 μM)) and a mixture of ParR (2.96 μM) and *parC* (251.56 ng/μL). Source data are provided as a Source Data file.

weight of ParM was estimated to be ~39 kDa (Fig. 2a), which is identical to that of the designed protein. The molecular weight of ParR (Fig. 2b) was ~14 kDa, which is consistent with previous results[39].

Biotinylated *parC* (230 bp) was obtained by using a biotinylated forward primer during the PCR process (Fig. 2c). To visualize gene segregation, biotinylated *parC* was attached to streptavidin-coated beads. Lane 2 (Fig. 2d) demonstrated that *parC* was successfully coated on the beads because the products remained close to the starting point, while the free *parC* was located at 230 bp. The biotinylated *parC*-beads were fluorescently labeled with SYBR Green I for subsequent visualization of DNA segregation (Supplementary Fig. 2).

ParM possesses ATPase properties that catalyze the hydrolysis of ATP to produce ADP and inorganic phosphate (Pi), which was subsequently used to estimate the ATPase activity of ParM. Pi release linearly increased with time in the first phase and then stabilized after 4 mM ATP was hydrolyzed by different concentrations of ParM (Supplementary Fig. 3). Pi release linearly increased as a function of ParM concentration within 1 h (Fig. 2e). The rate of ATP hydrolysation was estimated to be $16.58 \pm 0.01$ μM per μM (ParM) per hour, which was similar to the previously reported value of 16.2 μM per μM (ParM) per

hour[40]. ParR addition increased the ATPase activity of ParM (Fig. 2f) by ~1.47, 1.61, 1.62, 1.65, and 1.69-fold with ParR concentrations of 7.7, 23.0, 38.4, 76.8, and 115.1 μM, respectively.

Electrophoretic mobility shift assays were used to investigate the interaction between biotinylated *parC* and ParR. The unbonded *parC* signal disappeared gradually with increasing ParR concentration (orange rectangular box in Fig. 2g), while the *parC*-ParR complex signal was more pronounced (blue rectangular box in Fig. 2g), indicating that *parC* bonded to ParR. The peaks at 208 and 220 nm in the far-UV circular dichroism spectra corresponding to the alpha helix of ParR were not shifted after *parC* was added, indicating that *parC* did not influence the secondary structure of ParR (Fig. 2h).

## Polymerization behavior of ParM

In bacteria, DNA is segregated by ParM filaments[18], and the polymerization of ParM was found to occur in an ATP-dependent manner. ParM dots were generated with ATP concentrations less than 0.2 mM at a ParM concentration of 19.1 μM (Supplementary Fig. 4). The threshold ATP concentration for the formation of ParM filaments varied with ParM concentration (Fig. 3a), in which the blue dots

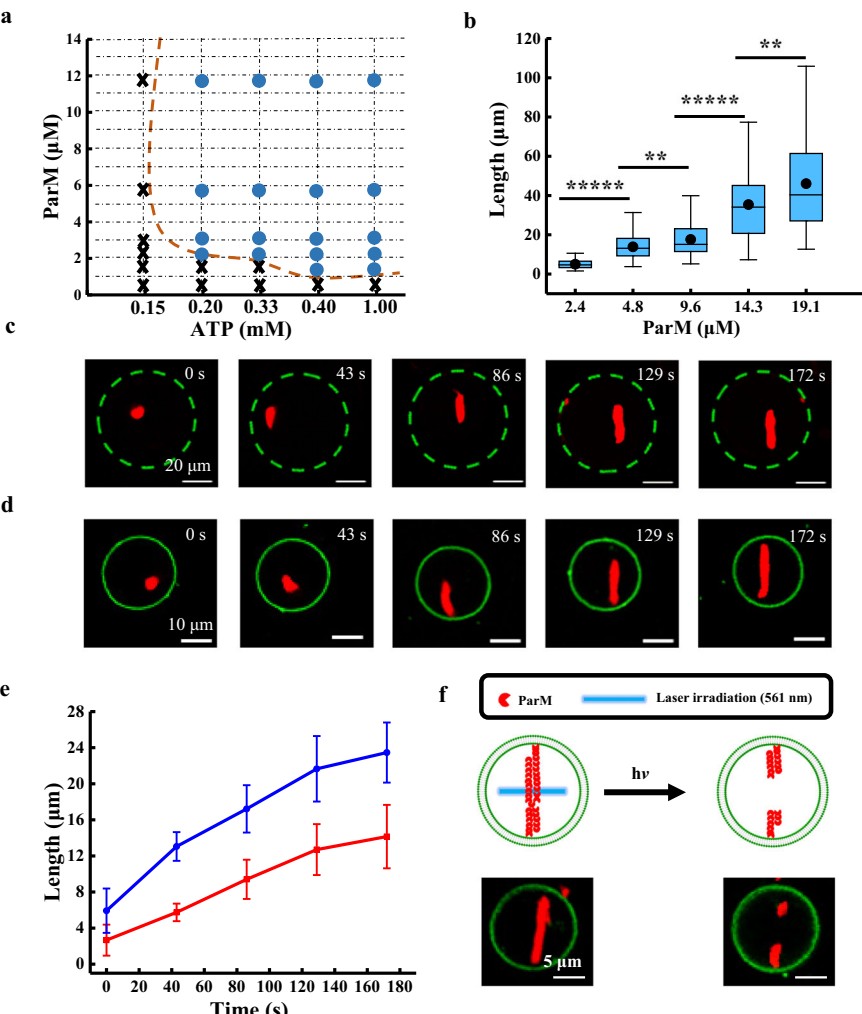

**Fig. 3 | Polymerization behavior of ParM. a** Phase diagram of ParM filament formation using ATP concentration (0.15, 0.2, 0.33, 0.4, 1 mM) and ParM concentration (0.6, 1.5, 2.2, 3.0, 5.9, 11.7 µM) as parameters. The cross symbols indicate that no ParM filaments formed under these conditions, while the blue dots indicate that ParM filaments could form under those conditions. **b** ParM filament length at ParM concentrations of 2.4, 4.8, 9.6, 14.3, and 19.1 µM. The concentration of ATP was 1 mM. The lengths of ParM were determined from 100 independent samples at each concentration. $n = 100$ independent ParM filaments. The central line corresponds to the median. The black dots correspond to the mean values. The lower and upper hinges of the boxes correspond to the 25th and 75th percentiles, respectively, and the whiskers represent the 1.5×interquartile range extending from the hinges. Statistical analyses were carried out by an unpaired two-tailed Student's *t*-test. **$P < 0.001$, *****$P < 0.000001$. The asterisks indicate significant differences of the lengths of filaments formed with ParM concentrations of 2.4 and 4.8 µM (Degrees of freedom = 198, $p < 0.000001$, effect size statistic = 0.6230, confidence intervals = 95%), ParM concentrations of 4.8 and 9.6 µM (Degrees of freedom = 198, $p = 0.000250$, effect size statistic = 1.013, confidence intervals = 95%), ParM concentrations of 9.6 and 14.3 µM (Degrees of freedom = 198, $p < 0.000001$, effect size

statistic = 1.865, confidence intervals = 95%) and ParM concentrations of 14.3 and 19.1 µM (Degrees of freedom = 198, $p = 0.000360$, effect size statistic = 2.938, confidence intervals = 95%). **c** Confocal microscopy images of a ParM filament as a function of time inside the lipid-protected droplet (green dotted circle for eyeguide) with a ParM concentration of 9.6 µM and an ATP concentration of 1 mM. The scale bars are 20 µm. **d** Confocal microscopy images of ParM filaments as a function of time inside the giant unilamellar vesicles (GUV) with a ParM concentration of 9.6 µM. ATP (1 mM) outside the GUV membrane flew inside upon 5 s of laser irradiation (405 nm). Scale bars are 10 µm. **e** The length of ParM filaments inside lipid-protected droplets (blue curve) and GUV (red curve) as a function of time. The length of ParM filaments inside the lipid-protected droplet (blue curve) and GUV (red curve) were obtained from three independent samples. The data were presented as the mean values ± SDs. $n = 3$ independent replicates. **f** Schematic and confocal microscopy images of ParM filaments before (left column images) and after (right column images) laser irradiation (561 nm, 0.7 mW, 5 s). The scale bars are 10 µm. $n = 3$ independent replicates. Source data are provided as a Source Data file.

indicate the successful formation of ParM filaments under these conditions and the cross symbols indicate the failure of ParM filament formation. The influence of the crowder on ParM polymerization was also investigated. The ParM filaments were shorter after treatment with 10 mM ATP, regardless of the presence of the crowder (0.4% methylcellulose) (Supplementary Fig. 5a, b). ParM filaments were thicker in the presence of methylcellulose (0.4%) (Supplementary Fig. 5c) at 1 mM ATP than in the absence of methylcellulose (0.4%) at 1 mM ATP (Supplementary Fig. 5d). In the following experiments, no crowder was used for the ParM polymerization.

The length distributions of ParM filaments at different ParM concentrations (2.4, 4.8, 9.6, 14.3, and 19.1 µM) were investigated. With increasing ParM concentration, the filament length became longer (Fig. 3b). The average lengths of the filaments were 5.0 ± 2.2, 13.9 ± 5.8, 17.6 ± 8.3, 35.4 ± 16.7, and 46.1 ± 24.2 µm at ParM concentrations of 2.4, 4.8, 9.6, 14.3, and 19.1 µM, respectively (Fig. 3b and Supplementary Fig. 6). The average filament length of 17.6 ± 8.3 µm at a ParM concentration of 9.6 µM fit the GUV size well (~20 µm in diameter; Supplementary Fig. 7). Therefore, 9.6 µM ParM was chosen for segregating DNA molecules inside the GUV.

To trigger the polymerization of ParM inside GUVs, ATP molecules were introduced into GUV from the outside upon light irradiation in the presence of Chlorin e6 (Ce6) according to the protocol described in the previous paper[35] (Supplementary Fig. 1). Ce6 is a commonly used photosensitizer that is isolated from Chlorella ellipsoidea[41]. Reactive oxygen species are produced by Ce6 under laser (405 nm) irradiation (Supplementary Fig. 8)[35,42], which causes unsaturated lipids to undergo peroxidation. This process affects membrane permeability because transient pores are formed in the lipid bilayer, allowing ATP to enter GUVs.

To determine the effect of Ce6 concentration and light strength of the laser on membrane permeability, the 8-hydroxypyrene-1,3,6-trisulfonic acid trisodium salt (HPTS) diffusion coefficient (D) across the lipid bilayer was investigated as a function of Ce6 concentration and laser intensity. The effect of Ce6 concentration (2, 4, 5, or 10 µM) on HPTS inflow rates was explored (Supplementary Fig. 9). The normalized fluorescence intensity measured from the corresponding images as a function of time with laser irradiation (405 nm, 0.5 mW, 5 s) is plotted in Supplementary Fig. 9. The D values were calculated to be $1.5 \times 10^{-11} \, \text{m}^2/\text{s}$, $2.5 \times 10^{-11} \, \text{m}^2/\text{s}$, $3.0 \times 10^{-11} \, \text{m}^2/\text{s}$, and $4.0 \times 10^{-11} \, \text{m}^2/\text{s}$, respectively (Supplementary Fig. 10). A higher Ce6 concentration resulted in faster diffusion of HPTS. Ce6 (10 µM) was chosen for the subsequent experiments to investigate the influences of laser intensity (Supplementary Fig. 11). The D values of HPTS as a function of laser power were measured in the presence of Ce6 (10 µM) upon laser irradiation for 5 s (Supplementary Fig. 12). Based on the normalized fluorescence intensity measured from the corresponding images as a function of time with laser intensity[43] (Supplementary Fig. 11), the diffusion coefficient was $0.2 \times 10^{-11}$, $1.7 \times 10^{-11}$, $2.3 \times 10^{-11}$, $2.7 \times 10^{-11}$, $5.3 \times 10^{-11}$, and $6.4 \times 10^{-11} \, \text{m}^2/\text{s}$, corresponding to laser intensities of 0.1, 0.3, 0.5, 0.7, and 0.9 mW, respectively (Supplementary Fig. 12). A faster penetration rate was observed with higher laser powers. ParM was found to polymerize adjacent to the lipid bilayer of GUVs with a laser power greater than 0.3 mW (Supplementary Fig. 13). Therefore, a laser power of 0.3 mW was chosen for the polymerization of ParM in the GUV system to segregate DNA.

To mimic the behavior of ParM polymerization in bacteria, we investigated the formation behavior of ParM filaments in lipid-protected droplets and GUVs. The polymerization buffer solution that contained ParM and ATP was added to mineral oil that contained POPC lipids to form lipid-protected water droplets by vortexing. The growth of ParM filaments was observed inside lipid-protected droplets from short filaments (5.9 ± 2.5 µm) at 0 s to long filaments (23.5 ± 3.3 µm) at 172 s (Fig. 3c and Supplementary Movie 1). The length of the ParM filament was plotted as a function of time (blue, Fig. 3e), from which an average growth rate of ~41 ± 7 monomers s$^{-1}$ was obtained, assuming a monomer length of 2.45 nm[19].

We then tried to form ParM filaments inside the GUV. In this case, the polymerization buffer that contained ParM was encapsulated inside the GUV. ATP was inflowed from external GUVs upon laser irradiation at 405 nm, and the filaments grew gradually inside the GUV (Fig. 3d and Supplementary Movie 2). From the growth curve (red curve in Fig. 3e), the elongation rate of the ParM filament was ~27 ± 4 monomers s$^{-1}$. In comparison, no ParM filaments inside the GUVs were observed in the absence of Ce6 upon the addition of 1 mM ATP and laser irradiation (Supplementary Fig. 14a), or in the presence of Ce6 and 1 mM ATP without laser irradiation (Supplementary Fig. 14b). The ability of the polymerized ParM filaments to segregate DNA towards two poles of GUVs was explored in subsequent experiments.

To divide the DNA in each pole of GUV into two daughter GUVs, it is essential to split the ParM filaments in the middle. In nature, parC DNA is pushed to opposite cell poles by ParM filaments with the help of ParR, followed by the detachment of ParR and parC DNA from ParM filaments; as a result, ParM-bound ATP underwent hydrolysis to depolymerize ParM filaments from their two ends[18]. However, the mechanism underlying ParR and parC DNA release from ParM filaments has not been investigated. Here, the ParM filaments were precisely cut into two parts, both in solution (Supplementary Fig. 15) and inside the GUV (Fig. 3f) after strong laser irradiation (561 nm, 0.7 mW, 5 s) according to the method described in previous work[19], due to the laser-induced photochemical reaction of Cy3 dyes which were attached to ParM via chemical bonds. Upon laser irradiation, Cy3 dyes generate free radicals, which damage ParM proteins and consequently depolymerize ParM filaments[44].

## DNA segregation driven by the ParMRC system inside the GUV

ParR and parC were introduced to mimic DNA segregation in bacteria (Fig. 4a). To visualize parC DNA segregation under a microscope, the parC DNAs were attached to beads (d = 1352 ± 206 nm). The green dots at the two ends of the filament were observed to separate gradually to reach 7.4 µm at 172 s in solution (Fig. 4b), which confirmed that the DNAs were segregated by ParM filament growth. These two dots are parC-beads attached at the ParM filament end via ParR. We further demonstrated successful DNA segregation via the ParMRC system inside GUVs (Fig. 4c). The ParMRC system was encapsulated inside GUVs. The ParM filament (Fig. 4a) started to grow after the inflow of external ATP upon laser irradiation (405 nm, 0.3 mW, 5 s). The two parC-beads (white arrows) were pushed towards the poles of the GUV (Fig. 4c) by a ParM filament at a rate of ~25 ± 5 monomers s$^{-1}$ (Fig. 4d). The parC DNA inside the GUV did not leak through the transient pores formed by Ce6 and laser irradiation (Supplementary Fig. 16). Thus, DNA segregation was successfully mimicked, which paved the way for mimicking the inheritance of genetic material after bacterial division.

## Bacterial plasmid segregation and inheritance mimicry using artificial cells containing the ParMRC system and PURE system

We demonstrated that the DNA was segregated by the ParMRC system (Fig. 4c) and that the ParM filament was split by laser irradiation (Fig. 3f). We further aimed to divide the artificial cells into daughter cells, which contain genetic information on the mother cell (Fig. 5a). GUV division was achieved using the following two-step procedure: first, the GUVs were deformed into a dumbbell shape (Supplementary Fig. 17) under hypertonic conditions, and second, Ce6-mediated peroxidation of unsaturated lipids was performed under laser (405 nm, 0.3 mW, 5 s) irradiation (Supplementary Fig. 18)[35]. Laser irradiation was focused on the central area of the dumbbell-shaped GUV, in which Ce6 molecules produced reactive oxygen species to oxidize local unsaturated lipids, resulting in a spontaneous increase in the membrane curvature and consequently splitting into two daughter GUVs. After the parC-beads were segregated at two poles of GUV by the splitting of the ParM filament, the artificial cells were divided into two daughter GUVs that contained genetic materials (parC-beads) (Supplementary Fig. 19).

To mimic genetic information inheritance, eGFP was chosen as a model protein[45] for expression inside daughter GUVs. Biotinylated parC-eGFP DNA was obtained by using a biotinylated forward primer during PCR (Supplementary Fig. 20). Its size is ~1000 bp, which is consistent with the theoretical value. The activity of parC-eGFP DNA was verified by the expression of eGFP in solution using the PURE system within ~4 h at 37 °C (Supplementary Fig. 21a). The expression of eGFP in solution was further confirmed by fluorescence spectrometry (Supplementary Fig. 21b). eGFP was also expressed inside the GUVs since the GUVs turned green gradually within ~4 h at 37 °C (Supplementary Fig. 22). The eGFP inside the GUV did not leak through the transient pores formed by Ce6 and laser irradiation (Supplementary Fig. 23).

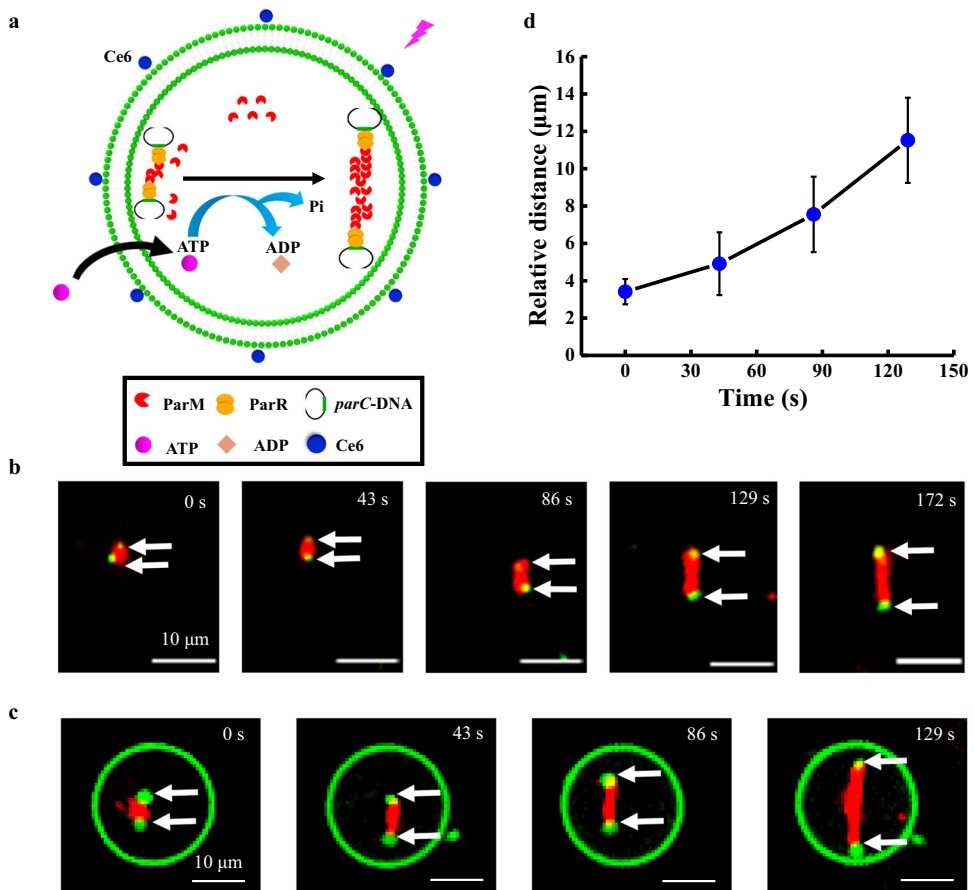

**Fig. 4 | Deoxyribonucleic acid (DNA) segregation driven by the ParMRC system inside the giant unilamellar vesicles (GUV). a** Schematic illustration of DNA segregation using the ParMRC system in GUV. ATP molecules entered GUV to trigger ParM polymerization upon laser irradiation (405 nm, 0.3 mW, 5 s). **b** Confocal time series images of the segregation of two *parC*-beads (green) through ParM (red fluorescence) polymerization in solution. The white arrows indicate the *parC*-beads. The solution contained ParM (9.6 μM), ParR (53.6 μM), green *parC*-beads, 1 mM ATP, 100 mM KCl, 30 mM Tris·HCl, 2 mM MgCl₂, and 1 mM Dithiothreitol (DTT). The scale bars are 5 μm. *n* = 3 independent replicates.

**c** Confocal time series images of two *parC*-beads segregated by ParM filament in a GUV that contained ParM (9.6 μM), ParR (53.6 μM), green *parC*-beads, 100 mM KCl, 30 mM Tris·HCl, 2 mM MgCl₂, and 1 mM DTT. ATP (1 mM) was inflowed from the outside upon laser irradiation (405 nm, 0.3 mW, 5 s). The scale bars are 10 μm. *n* = 3 independent replicates. **d** Time-dependent relative distance between two *parC*-beads as a function of time. The distance of two *parC*-beads s inside the GUV were obtained from three independent samples. The data are presented as the mean values ± SDs; *n* = 3 independent replicates. Source data are provided as a Source Data file.

The full mimicry of bacterial plasmid segregation and inheritance was completed using artificial cells that contained the ParMRC system and PURE system (Fig. 5c). The biotinylated *parC*-eGFP was prepared using a method similar to that used for biotinylated *parC*. The amount of biotinylated *parC*-eGFP DNA on each bead was estimated to be ~0.87 ng, which was enough to trigger eGFP expression using PURE systems. The *parC*-eGFP DNAs were subsequently pushed to the two poles of the GUV by ParM polymerization. The GUV was deformed by hypertonic solution (△c = 70 mM) (Fig. 5b1), followed by ParM filament splitting upon laser irradiation (561 nm, 0.7 mW, 5 s) (Fig. 5b2) and GUV division by laser irradiation (405 nm, 0.5 mW, 5 s) (Fig. 5b3). The *parC*-eGFP DNA in the two daughter GUVs was translated into enhanced green fluorescent proteins using the PURE system at 37 °C (from Fig. 5b4 to b7). The fluorescence intensity of eGFP expressed in the two daughter GUVs as a function of time was monitored (Fig. 5d, e). Negligible green fluorescence was observed within 30 min; afterward, the fluorescence intensity inside daughter cells 1 and 2 increased gradually and levelled off at nearly 300 min. The degree of eGFP expression inside daughter GUV 1 and 2 was similar. In comparison, no green fluorescence was observed at 37 °C for 3 h inside daughter cells when the mother cells did not contain the PURE system (Supplementary

Fig. 24). Without the ParMRC system, the *parC*-eGFP DNA beads were randomly distributed into two daughter GUVs (Supplementary Fig. 25), which resulted in a significant difference in the expression rate between the two daughter GUVs (Supplementary Fig. 26). In contrast, the expression rates of each daughter GUV with the ParMRC system were almost identical (Supplementary Fig. 26). Without the ParMRC system, only 1 GUV was divided into daughter GUVs with one *parC*-eGFP DNA bead in each daughter GUV out of 11 GUVs that divided into equally sized daughter GUVs. The bead segregation rate is ~9% without the ParMRC system. This may be caused by gravity and weak magnetic interaction of two magnetic *parC*-eGFP DNA beads. The asymmetric division also exists in real cells without active segregation systems. Mitochondria in eukaryotic cells were equally distributed to daughter cells by actin cables at cytokinesis; however, asymmetrical allocation of mitochondrial mass between daughter cells happened with the absence of actin cables[46,47]. With the ParMRC system, the bead segregation rate is ~82% (*n* = 11). The successful bead segregation rate is much higher in the GUVs containing the ParMRC system than those containing no ParMRC system, which further confirmed the role of the ParMRC system in segregating genetic information in both daughter GUVs.

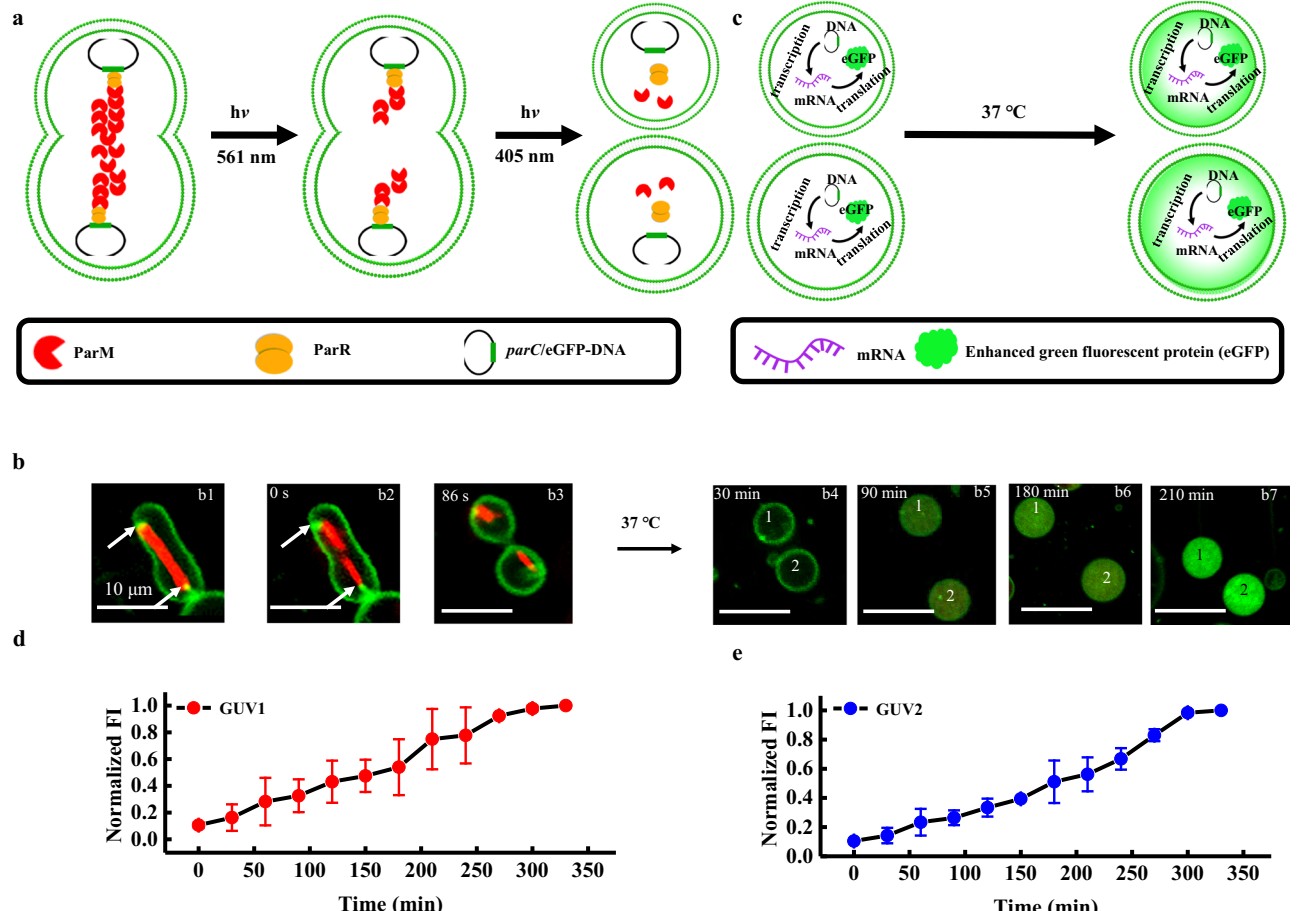

**Fig. 5 | Bacterial plasmid segregation and inheritance mimicry using artificial cells containing the ParMRC system and protein synthesis using recombinant elements (PURE) system. a** Schematic illustration of two *parC*-eGFP DNAs being pushed through ParM polymerization in a giant unilamellar vesicles (GUV) and subsequent GUV division under laser irradiation. **b** Confocal microscopy images of GUV division (**b1**–**b3**) and enhanced green fluorescent protein (eGFP) expression at 37 °C (**b4**–**b7**). **b1**–**b3** indicate GUV deformation, filament splitting, and division into two daughter cells, respectively. The white arrows in **b1** and **b2** indicate *parC*-

eGFP DNA. The ParM filament was split by laser irradiation (561 nm, 0.7 mW, 5 s). The scale bars are 10 μm. *n* = 3 independent replicates. **c** Schematic illustration of eGFP expression in two daughter cells. The corresponding fluorescence intensity (FI) in daughter GUV 1 (**d**) and daughter GUV 2 (**e**) as a function of time. The normalized fluorescence intensity of eGFP was calculated from three independent samples. The data were presented as the mean values ± SDs; *n* = 3 independent replicates. Source data are provided as a Source Data file.

## Discussion

To mimic bacterial plasmid segregation and inheritance, the ParMRC system was successfully purified and reconstituted into GUVs to form artificial cells. DNA was segregated into two poles of the GUV by ParM filaments with the help of ParR and *parC*. ParM polymerization was triggered by ATP molecules, which inflowed from outside the GUV upon laser irradiation in the presence of Ce6. GUV division was achieved by a two-step protocol, i.e., deforming the cells into a dumbbell shape by hypotonic conditions, followed by ParM filament splitting and division upon laser irradiation. The eGFP gene in daughter cells, a genetic material inherited from mother cells, was translated into eGFP via the introduction of the PURE system. We developed an artificial cell that can dynamically mimic the segregation process of plasmids with low copy numbers as well as the gene inheritance of bacteria after division. This paper paves the way for the construction of more sophisticated artificial cells that can mimic prokaryotic division.

## Methods

### Materials

1-Palmitoyl-2-oleoyl-glycero-3-phosphocholine (POPC) was obtained from Avanti Polar Lipids (USA). N-(7-Nitrobenz-2-oxa-1,3-diazol-4-yl)-1,2-dihex-adecanoyl-*sn*-glycero-3-phosphoethanolamine

triethylammonium salt (NBD-PE) was obtained from Thermal Fisher Scientific (USA). Cy3 maleimide (nonsulfonated) was purchased from APExBIO (USA). A Bradford protein assay kit, SDS–PAGE gel configuration kit, 4×SDS–PAGE sample loading buffer, protease inhibitor cocktail for purification of His-tagged proteins, green fluorescent DNA marker dye, streptavidin magnetic beads, Tris-borate-EDTA buffer (TBE) and Tween-20 were purchased from Beyotime (China). Protein markers (14.4−-97.4 kDa), 2×SYBR Green PCR Master Mix, a Na⁺K⁺-ATPase assay kit, TE buffer (pH 8.0), deoxyribonuclease I (DNase I), deoxyribonucleic acid sodium salt from salmon testes, and dithiothreitol (DTT) were obtained from Solarbio (China). A Ni-NTA 6FF prepacked chromatographic column and DNA marker (100−5000 bp) were purchased from Sangon Biotech (China). The plasmid pET28a containing the ParM gene, the plasmid pET28a containing the ParR gene, the plasmid pBR322 containing the *parC* gene, the forward primer (primer 1) and the reverse primer (primer 2) of the *parC* gene, and the forward primer (primer 1′) and the reverse primer (primer 2′) of the *parC*-eGFP gene were obtained from Sangon Biotech (China). Yeast extract fermentation agent, tryptone, and agar were purchased from AOBOX (China). Lysozyme, isopropyl β-D-thiogalactoside (IPTG), and imidazole were purchased from Biotopped (China). Sucrose, glucose, magnesium chloride (MgCl₂), sodium chloride (NaCl), adenosine 5′-

triphosphate disodium salt (ATP), 8-hydroxypyrene-1,3,6-trisulfonic acid, methylcellulose, trisodium salt (HPTS) and bovine serum albumin (BSA) were purchased from Sigma (China). Phosphate-buffered saline (PBS) and trypsin-ethylenediaminetetraacetic acid (EDTA) were purchased from Corning (USA). Chlorine e6 (Ce6) was purchased from MREDA (USA). Millipore Milli-Q water with a resistivity of 18.2 MΩ·cm was used in the experiments.

## Instruments
A gel imaging instrument (Amersham Imager 600, GE) was used to detect the bands of the extracted ParM, ParR, and *parC*. A microplate reader (Molecular Devices, SpectraMax iD3, Germany) was used to determine the protein concentration and phosphate release. A circular dichroism instrument (Chirascan, Applied Photophysics, UK) was used to analyse the secondary structure of ParR and ParR-*parC*. All fluorescence images were obtained with a fluorescence microscope (Olympus IX73, Japan) and a confocal laser scanning microscope (Olympus FV 3000, Japan). A UV-Vis spectrometer (Agilent Technologies Cary 60) was used to measure the concentration of ATP. A fluorescence spectrometer (PerkinElmer Fluorescence spectrometer LS 55) was used to measure the fluorescence emission spectrum of the eGFP synthesized in vitro. An automated cell counter (Countess 3, Thermo Fisher Scientific) was used to measure the concentration of the streptavidin beads.

## Expression and purification of ParM and ParR
The ParM gene was synthesized, cloned, and inserted into the expression vector pET28a (a plasmid encoding an N-terminal 6-histidine tag) to generate the ParM plasmid (Supplementary Data 1), which was subsequently transformed into *E. coli* BL21 (DE3) cells at a certain concentration (-10 ng). After the cells were grown to an OD600 of -0.8, 1 mM IPTG was added at 15 °C overnight to induce the expression of ParM. The cell pellets were collected by centrifugation at $2500 \times g$ for 5 min and resuspended in a mixture of 50 mM Tris-HCl, 500 mM NaCl, 20 mM imidazole, 0.2 mg/ml lysozyme, protease inhibitor (100X), 0.02 mg/ml DNase I, and 1 mM $MgCl_2$ at pH 8.0. After lysing, the cell lysate supernatant was collected via centrifugation at $15,000 \times g$ for 30 min and loaded onto a Ni-NTA 6FF prepacked chromatographic column that was preequilibrated with 50 mM Tris-HCl, 500 mM NaCl, and 20 mM imidazole at pH 8.0. The ParM was purified by eluting with a series of buffer solutions containing different concentrations (50, 100, 200, 300, 400, and 500 mM) of imidazole. The ParM in different fractions was analysed via SDS–PAGE. The relatively pure ParM fraction (200 mM imidazole) was further purified by a size-exclusion chromatographic column in 260 mM sucrose, 30 mM Tris-HCl, 2 mM $MgCl_2$, 1 mM DTT, and 100 mM KCl at pH 7.5. The obtained ParM was concentrated and stored at −80 °C. The ParR gene was synthesized, cloned, and inserted into the expression vector pET28a to obtain the ParR plasmid (Supplementary Data 1). ParR was extracted using a method similar to that used for ParM. The purities of ParM and ParR were greater than 99%, as determined by densitometric analysis with ImageJ. The N-terminal 6-histidine tag was not cleaved before use. The extracted proteins were stored at −80 °C.

## Labeling of ParM with fluorescent dyes
Five additional amino acids (GSKCK) were added to the C-terminus of the ParM gene for fluorescent labeling of the Cy3 maleimide dye. Cy3 maleimide dye (16 μM) was mixed with ParM solution (96 μM) for 10 min at 25 °C, followed by the addition of 10 mM DTT to quench the reaction. The free Cy3 maleimide dye molecules were removed by centrifugation in an ultrafiltration tube (10 kDa) at $15,000 \times g$ for 20 min in a mixture of 260 mM sucrose, 30 mM Tris-HCl, 2 mM $MgCl_2$, 1 mM DTT, and 100 mM KCl at pH 7.5 three times. The samples were stored at −80 °C.

## Coverslip preparation
The coverslips were cleaned by ultrasonication in dichloromethane for 15 min, dried under a stream of nitrogen, and rinsed in Milli-Q water before being immersed in piranha solution (70:30, v/v, $H_2SO_4$:$H_2O_2$) for 5 min. The coverslips were subsequently washed with Milli-Q water and ethanol. The coverslips were stored in ethanol until use. The polymerization of ParM and subsequent *parC*-beads segregation in solution were performed using these treated coverslips.

## Splitting of ParM filaments by laser irradiation
ParM filaments were precisely cut into two parts, both in solution and inside the GUVs, using high-intensity laser irradiation (561 nm, 0.7 mW, 5 s) through laser scanning at the central area (-1 μm in diameter) of the filaments several times until full splitting was achieved. The laser intensity was measured by an optical densitometer (CEL-NP2000).

## Preparation of biotinylated *parC*
The biotinylated forwards primer (5′-TTCCATATGTTGTTACCCGCC AAACAAAACCCA-3′) and the reverse primer (5′-CGGGATCCAGTTT GATGTTTGTTGAACCGTCATC-3′) were used to amplify the biotinylated *parC* gene (Supplementary Data 1) via the classical PCR technique. Specifically, a 50 μL PCR mixture [2×SYBR Green PCR mix (25 μL), primer 1 (10 μM, 5 μL), primer 2 (10 μM, 5 μL), template DNA from the plasmid pBR322 *parC* (10 μL), and ddH₂O (5 μL)] was subjected to thermal cycling (5, 10, 15, 20, 30, and 40 cycles) under the following thermal conditions: 95 °C for 10 min, [95 °C for 20 s, 60 °C for 30 s, and 72 °C for 60 s]. Biotinylated *parC* was obtained and stored at 4 °C (Supplementary Fig. 27a, b). The biotinylated eGFP-*parC* DNA was prepared using a method similar to that used for biotinylated *parC*.

## Preparation of beads modified with *parC*
Streptavidin magnetic beads were concentrated on a magnetic separation rack for 1 min and then washed three times with TBS buffer solution (20 mM Tris-HCl, 0.137 M NaCl, pH 7.5) by the magnetic separation method. Afterward, the streptavidin magnetic beads were resuspended in 200 μL of biotinylated *parC* solution for 30 min. The free biotinylated *parC* was removed by magnetic separation in a mixture of 10 mM Tris-HCl, 1 mM EDTA, 2 M NaCl, and 0.05% Tween-20, pH 7.5, three times (Supplementary Fig. 28a, b). The sample was resuspended in 50 μL of buffer solution (260 mM sucrose, 30 mM Tris-HCl, 2 mM $MgCl_2$, 1 mM DTT, 100 mM KCl, pH 7.5) and stored at 4 °C. The size of the *parC*-beads was $1352 \pm 206$ nm according to the DLS data (Supplementary Fig. 29). The number of plasmids on each bead was estimated using the following calculations. The concentration of streptavidin beads was $(1.653 \pm 0.006) \times 10^6$/mL using an automated cell counter. The concentration of biotinylated *parC* DNA was $466 \pm 24$ ng/μL, as determined via UV–vis spectroscopy. The mixture of streptavidin beads and biotinylated *parC* DNA was incubated for 30 min at room temperature. After magnetic separation, the supernatant was removed and the *parC* DNA concentration was determined to be $419 \pm 1$ ng/μL. The *parC*-beads were washed again using the same protocol. The *parC* DNA concentration in the supernatant was $1.17 \pm 0.29$ ng/μL. After the third wash, the concentration of *parC* DNA in the supernatant was 0. Thus, the total amount of plasmid bound to the beads was calculated to be 72191.5 ng by subtracting the remaining plasmid DNA in the supernatants (419 ng/μL × 50 μL + 1.17 ng/μL × 50 μL) from the added plasmid DNA (466 ng/μL × 200 μL). The average amount of *parC* DNA was -0.87 ng when 72191.5 ng was divided over 82,650. This amount of DNA single beads was sufficient to trigger eGFP expression using PURE systems.

## ParM ATPase activity assay
The ATPase assay for ParM was performed in accordance with the kit protocol. Under strong acid conditions, the phosphates

released from ATP hydrolyzed by ParM react with ammonium molybdate to produce phosphomolybdate yellows, which are reduced by stannous oxide (SnO) to phosphomolybdate blue. The intensity at 660 nm of the solution is proportional to the phosphate concentration. The ATPase activity of ParM was calculated using the following Eq. 1:

$$ATPase \ activity \ of \ ParM = \frac{C_{standard} * V_{total} * (A_{experiment} - A_{control})}{T * (A_{standard} - A_{blank}) * (C_{ParM} * V_{sample})} \tag{1}$$

where $A_{experiment}$, $A_{control}$, $A_{standard}$, and $A_{blank}$ represent the absorbance at 660 nm of the experimental group, control group, standard sample, and blank sample, respectively, with a $C_{standard}$ of 0.5 μmol/mL, a $V_{total}$ of 0.25 mL, a $V_{sample}$ of 0.1 mL, and a $T$ of 10 min.

### Electrophoretic mobility shift assays
*parC* (275 ng) was mixed with ParR (0, 24, 48, 140, 240, 480, and 720 μM) in a buffer solution (20 mM HEPES, 150 mM KCl, 5 mM MgCl₂, 1 mM DTT, 1 mg/ml BSA, 0.1 μg/μL sonicated salmon sperm DNA, 5% glycerol, pH 7.6). These samples were incubated at room temperature for 30 min and then loaded on a 1.5% magnesium-agarose gel. Electrophoresis was conducted at 150 V for 2 h using running buffer (45 mM Tris-HCl, 45 mM boric acid, and 5 mM magnesium acetate). The results were recorded using an Amersham Imager 600 (GE).

### Circular dichroism (CD) measurements
ParR (3.31 μM) and a mixture of ParR (2.96 μM) and *parC* (251.56 ng/μL) were analysed using a circular dichroism instrument. The spectra were recorded with 1 nm steps and a dwell time of 2 s per step in a wavelength range of 180–300 nm.

### Preparation of ParM-containing GUVs
Giant unilamellar vesicles (GUVs) were fabricated by a water-in-oil (w/o) emulsion-transfer method[48]. Specifically, 1 mg of POPC and 0.05 mg of NBD-PE were dissolved in 2 mL of mineral oil (lipid-mineral oil solution). The aqueous solution contained 260 mM sucrose, 30 mM Tris-HCl, 9.6 μM ParM, 1 mM ATP (if necessary), 2 mM MgCl₂, 1 mM DTT, and 100 mM KCl, pH 7.5. Then, 30 μL of aqueous solution was mixed with 300 μL of lipid-mineral oil solution by vortexing for 30 s to generate a water-in-oil emulsion solution. The emulsion solution was gently added to 200 μL of isotonic glucose solution in a 1.5 mL centrifuge tube, followed by centrifugation (10,000 × g, 30 min) at 4 °C. The GUVs were collected at the bottom of the tube and used for subsequent experiments. ParMRC-containing GUVs were prepared through a similar method as that used for the ParM-containing GUVs. Since the amount of eGFP plasmid on each bead (~0.87 ng) is sufficient for expression in GUVs containing the PURE system, we encapsulated two *parC*-beads in each GUV as much as possible; as a result, the beads bound to the two ends of ParM filaments, avoiding the free unbonded beads in the GUV.

### Ce6-mediated HPTS influx
GUVs (containing 260 mM sucrose) were mixed with 1 mM HPTS (containing 260 mM glucose) supplemented with Ce6 at 1.8, 3.6, 5.3, or 10.4 μM. These samples were treated by laser irradiation at 405 nm with light intensities of 0, 0.1, 0.3, 0.5, 0.7, and 0.9 mW for 5 s. Fluorescence microscopy images of the HPTS inside the GUV were recorded by confocal laser scanning microscopy (Olympus FV 3000, Japan) as a function of time.

### Ce6-mediated GUV division
GUVs (containing 260 mM sucrose) were mixed with 400 mM glucose solution to induce the deformation of the GUVs. Once GUVs were prolonged due to the osmotic pressure, 10 μM Ce6 was added to achieve GUV division by laser irradiation at 405 nm for 5 s.

### Gene expression of enhanced green fluorescence protein (eGFP)
A total of 20 μL of the PUREfrex 2.1 system containing 8 μL of solution I (amino acids, NTPs, tRNAs, enzyme substrates, etc.), 1 μL of solution II (proteins), 2 μL of solution III (20 μM ribosomes), 2 μL of cysteine (3 mM), 1 μL of GSH (80 mM), 1 μL of the plasmid encoding eGFP (20 ng/μL), and 5 μL of nuclease-free water was encapsulated in the GUV by the emulsion method. After the GUVs were incubated at 37 °C, the expression of eGFP was observed by fluorescence microscopy as a function of time.

### Reporting summary
Further information on research design is available in the Nature Portfolio Reporting Summary linked to this article.

## Data availability
All data generated in this study are provided in the Supplementary Information and Source Data files. Source data are provided with this paper.

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

## Acknowledgements

This work was supported by the National Natural Science Foundation of China (Nos. 22174031, 22374033, and 21929401 to X.H.), the Natural Science Foundation of Heilongjiang Province (No. ZD2022B001 to X.H.), and the Heilongjiang Touyan Team Project (No. HITTY20190034 to X.H.)

## Author contributions

X.H. supervised the research. X.H. and J.Z. conceived and designed the experiments. J.Z. performed experiments. X.H. and J.Z. analyzed the data and wrote the paper.

## Competing interests

The authors declare no competing interests.
