## [Peer Review File · Nature Communications]

Reviewers' Comments:

Reviewer #1:

Remarks to the Author:

The manuscript by Han et al. demonstrate the reconstitution of a DNA segregation machinery into GUVs followed by GUV division. This is an impressive achievement. I see it as one of the most complex systems that have been reconstituted in bottom-up synthetic biology. The images speak for themselves, but in some parts the writing could be clearer. I have a few comments below, which should be addressed before publication.

Comments:

1. Storyline: Reading the beginning of the results section, I was worried that the paper is very superficial. But indeed, the data follows. It just has to be clear for the reader at the beginning of the results section that the first part describes the whole concept and the results will be presented later.
2. Have the authors performed the controls for Ce6-mediated influx (addition of ATP without Ce6, addition of ATP and Ce6 without illumination)? ATP influx could also be confirmed by using fluorescent ATP or with a luciferase assay (no need to do this if the controls above are successfully performed).
3. Is it really possible to split the ParM with 560nm? Have the authors tested this in bulk outside of the GUV? What is the mechanism? The authors link to Ref 4, but a bit more explanation in the manuscript text would be nice.
4. It would also be nice to add in the discussion a paragraph about how ParM splitting could be achieved autonomously without laser irradiation.
5. Figure 2 e, f: How was phosphate release and ATPase activity measured? What are the error bars? How often was this repeated?
6. The authors study polymerization of ParM in droplets and GUVs (Figure 3c, d). Is it true that for the droplets the ParM tends to localize at the periphery more? This is how it looks in the image. Do the authors have any idea why? Shouldn't that "consume" monomers such that the polymerization rate in droplets is expected to be slower compared to GUVs?
7. "In order to visualize parC DNA segregation under microscope, they were attached onto beads (d=1352 nm)." How was the bead size measured so accurately? Use appropriate number of significant digits.
8. "The deformation rate of GUVs was ~ 30 % (Figure S14b). The division rate of deformed GUVs was ~36 % after laser irradiation." I assume the authors mean the fraction of deformed or divided GUVs. It does not make sense to talk about a rate here.

Reviewer #2:

Remarks to the Author:

The authors describe the reconstitution of the bacterial ParMRC system in GUVs to drive localisation of plasmid DNA such that subsequent GUV division creates two daughter vesicles each containing the plasmid DNA. This represents minimal model artificial cell system for genetic transfer to daughter cells in artificial cell division.

ParM protein undergoes ATP dependent polymerisation into filamentous structures in the presence of the protein ParR that caps the filament end when bound to partner parC. ParC represents a DNA sequence within the target plasmid, thus anchoring the plasmid DNA to the poles of the filament. This assembly stabilises the growing filament with filament polymerisation thought to drive localisation of bacterial plasmid DNA to either end of a dividing cell, ensuring plasmid copies in both the resulting daughter cells. The ParMRC system is probably the most well characterised prokaryotic segregation machinery and function has also been widely demonstrated in vitro. The authors build on this underpinning work reconstituting the ParMRC system into GUVs and employing osmotic and chemical/light mediated mechanisms to drive vesicle breakup, creating an artificial cell system capable of DNA transfer into two daughter cells on forced division, analogous

to the principles of the original bacterial system.

The authors demonstrate success by GFP plasmid localisation and expression by IVTT in both compartments following vesicle division. However, this reviewer feels additional comparative controls and further experimental details/characterisation are required before publication. However, the overall concept pushes beyond the current literature and the controlled molecular machinery mediated DNA segregation in artificial cells with division will be of broad interest and represents a significant achievement. This is however realised by combining two previously established protocols a) for in vitro reconstitution of ParMRC and b) controlled vesicle division, so one may argue the outcome is fairly obvious and a logical next step. In this context, the background literature in both the above areas should be better covered in the introduction to make the work's place in the field more clear. In addition, some scientific questions remain and the manuscript requires some extensive improvements to the text before the work is ready for publication, including clarification of methodology as it is often hard to interpret exactly what was done.

Major comments:

Appreciating English may not be the author's first language, there are places where the work is hard to interpret due to poor grammar or sentence construction combined with lack of detail. This should be improved throughout. It may be that some of the following questions are in fact addressed in the data, but this reviewer could not unpick some the experimental results and methodological descriptions to understand exactly what was done and what was being reported at times.

A significant piece of prior work by the groups of Petra Schwille and Victor Sourjik should be acknowledged and described [ChemBioChem 2019, 20,2633–2642]. Whilst other works from these research groups are cited, this work doesn't appear to be. In this paper the researchers successfully reconstitute the ParMRC system with plasmid replication, using the T7 system to replicate circular plasmids with the Cre-lox system to regenerate them, and the ParMRC system for DNA segregation in vitro. ParMRC segregation is also demonstrated in biomimetic compartments driving plasmid localisation to the poles. Whilst these compartments are not engineered to undergo division, the underpinnings are highly relevant to the work under review and this represents important prior work that should be described and used to contextualise the advance being reported by the authors in the reviewed manuscript.

Vesicle division is an important element of the reported work, but only three or four lines are dedicated to this in the background. Whilst relevant literature is cited, there is no contextual detail for the reader. It would be appropriate to expand with some description of the mechanisms employed, how they work and results achieved - especially given the broad readership of nature communications who may not be familiar. The combination of vesicle dividing systems with DNA redistribution is only faintly touched upon "Some divisions were accompanied with DNA redistribution^{18,20,25,26}." This should really be expanded upon to give a fair assessment of the current state of the art, briefly summarising what approaches have been employed, what results they have delivered and associated strengths or drawbacks. This would allow the reader to understand the place of the submitted work and the advantages of controlled localisation via ParMRC over other approaches.

Detail on the Ce6 system should be added. This is fundamental to the study both in the import of ATP into the GUVs and it would seem the subsequent GUV division. However there is no

explanation of what Ce6 is or how it works. "...upon light irradiation with the help of Ce6 molecules and osmotic pressure change." is not a sufficient explanation for the mechanism of GUV division employed. Similarly "ATP molecules enter into GUVs through the transient pores due to the existence of Ce6." Does not adequately describe to the reader the mechanism/role of Ce6 in the system for ATP influx into the GUV.

How does the laser bring about splitting of the ParM filament? This is just described as "The ParM filaments was splited by laser (561 nm, 0.7 mW, 5 seconds) at center region of dumbbell GUV". Is the laser illumination wide-field and the ParM just happens to split in the middle? Or is a method of laser scanning employed to specifically target the centre of the ParM filament? More methodological details are required to understand what was done and appreciate why the ParM filament splits. Has this previously been reported? Later it is described as "A strong laser irradiation (561 nm, 0.7 mW, 5s) precisely cut ParM filament into two parts inside GUVs (Figure 3f), because the laser induced photochemical reaction⁴ of cy3 modified onto ParM." This is not clear and there is no insight provided into the mechanism. It is not obvious how ref 4 relates to this and it should be noted that biomolecules tagged with the fluorophore cy3, which is excited at 561nm are often exposed to powers far in excess of 0.7 mW, e.g. in single molecule imaging, without apparent reaction beyond the expected fluorescence processes. Does this higher laser power of 0.7 mW pose any problems with driving the Par system (or the beads?) to the GUV membrane given the manuscript states that laser powers above 0.3 mW were seen to to drive association of Par to the membrane (see later comment on this also).

The most important experimental questions and comments concern the final figure and the expression of eGFP following ParMRC segregation. The production of parC-eGFP coated beads is not described. Presumably this is the same as the earlier parC beads? This is essential to know given the reliance on the eGFP expression data to evidence the main claim of the manuscript, that of controlled DNA segregation by the ParMRC system prior to artificial cell division and subsequent gene expression. The reader has to assume what was done, which is not desirable for any presented data, especially not the key result. In this context there are several important questions that are not obviously answered by the controls and data/text presented concerning the eGFP expression experiments.

1) Is there any chance of free eGFP plasmid or unbound parC-eGFP coated beads being present inside the GUV and ending up in the two daughters simply by diffusion? Have control experiments without the ParM polymerisation been conducted to see what proportion of daughter vesicles are able to express eGFP after division with no active segregation? If this is possible, then how do we know for certain the presented result is (entirely) due to the ParMRC system? Comparing to the outcome with and without ParM(RC) would confirm the role of the Par system, where one would expect to observe an increase in probability of both daughters expressing eGFP with the ParMRC system. This comparison would give confidence in confirming the necessity of ParMRC to achieve the reported result of expression in both daughters. It would also allow quantitative comparison of the efficiency increase of the process due to ParMRC over random chance. To assist with understanding here the authors could estimate how many parC-eGFP coated beads they expect per vesicle. Will these likely all be bound to ParM filaments, or would some be expected to be free? Would imaging reveal the bound efficiency here? The number of anticipated plasmids per bead could also be estimated and correlated to expression rate. The presence or absence of free parC-eGFP plasmid could also easily be confirmed experimentally to rule out equilibration of free plasmid in the vesicle prior to division.

2) In Fig5b are the contents of the two vesicles definitely separated given the two vesicles are still in contact for at least 90 mins at the elevated temperature of 37 degrees used to initiate IVTT protein expression. Given the presence of Ce6 and its use to make pores in the membrane earlier in the protocol could plasmid or eGFP be exchanged via pores? Similarly in images B6 and B7 it looks the two vesicles are linked by a membrane tether. Could the two vesicles shown still be in diffusive contact? How repeatable was this result as only one example is shown in the manuscript.

Was this experiment successful more than once? In the example shown, in b3 is there a large third vesicle in the top left had corner of the image that the original vesicle(s) seem attached to? What is happening here?

3) line 264 "no green fluorescence was observed at 37°C for 3 h inside daughter cells when the mother cell did contain PURE system (Figure S20)." - presumably this should say did not?

Minor comments:

The nature of the ParMRC system could be better described for the non-expert, as understanding of this is essential for understanding the reported experiments.

The grammar in the introduction makes it not always clear whether the work of the paper or the work of others is being described. This can be worked deduced, but improved construction would help the reader.

Figure 1 is described and presented as if it were an experimental result, but is just a cartoon schematic. Discussing as a result at this stage of the manuscript makes the flow feel muddled when the subsequent data is gel purification. A schematic is a good idea for figure one but the text around it could be improved to help the reader and the flow of the manuscript.

Figure 2f - what is the 'control' condition of the black bars. Not clear from the text or legend label of just control.

In characterising the polymerisation behavior of ParM, why not represent all measured lengths in a histogram rather than categorising into arbitrary categories of different size ranges?

The text states "Ce6 incorporated spontaneously into the outer leaflet of GUV membrane (Figure S8)" It is not possible to tell from a microscopy image (Figure S8) of a vesicle whether the Ce6 has incorporated in the outer, inner or both leaflets. The membrane thickness is far smaller than the diffraction limit and the increased observed annulus thickness of the GUV is most likely a focus effect or optical artefact.

The text states "In order to determine the Ce6 concentration and light strength of laser, the 8-Hydroxypyrene-1,3,6-trisulfonic acid trisodium salt (HPTS) diffusion coefficient (D) across the lipid bilayer was investigated as a function of Ce6 concentration and laser intensity." This measurement does not report on laser power or Ce6 concentration, but on the effect of these two variables on the membrane.

In the same section "Polymerization behavior of ParM" In the discussion of the rationale behind FigS12 and S13, the authors are showing and discussing a range of experiments which were designed to inform the most appropriate laser power at which to carry into the next phase of the work - e.g. HPTS diffusion at varying laser powers is used as a measure of extent of pore formation, why not show the experimental results on which this decision on laser power was actually made as it is stated "ParM was found to polymerize adjacent to the lipid bilayer of GUVs with the laser power over 0.3 mW"? Which means most of the tested laser powers were ruled out. A lot of text in the main manuscript and data in the SI is dedicated characterising the relationship between laser power and HPTS diffusion, but no data is given to support the decision that was actually made.

Figure 3. It would be helpful to label c) as droplet and d) as vesicles. The text discussing the results of Fig3 states that filaments inside GUVs were smaller than in lipid coated droplets- can this really be confidently be concluded from the data of Fig3e with such large variance and just three replicates (n=3)?

Can the bead diameter really be measured as accurately as $d=1352$ nm?

Throughout it is not always totally clear what we are looking at in the microscopy images as the descriptions are not complete.

In FigS14, what constituted deformation in the assessment employed?

In. FigS20. The figure is titled "Confocal microscopy images of eGFP expression in two daughter cells without PURE system." this should be revised as there is no expression system and so no expression.

There are further examples of unclear statements in the main text and figure captions where clarification would show off the authors work a lot better and save the reader having to wrestle so much with the information to really understand the work.

Reviewer #3:

Remarks to the Author:

In their generally intriguing attempt towards mimicking synthetic cell division, Zhao and Han aim to combine the bacterial ParMRC segregation system with the PURE system to couple controlled genetic inheritance with vesicle splitting and protein expression in daughter vesicles. The objective to segregate DNA in a controlled way into dividing vesicles is timely and promising. However, the execution is unsatisfying, as the authors do not provide evidence for a significant role of the ParMRC system in the distribution of the DNA (eGFP gene) to offspring cells, leave alone a functional coupling to the (forced) division, as would be a truly noteworthy accomplishment. Furthermore, there are major deficiencies in the quality of the produced ParM and discrepancies between its behaviour shown here and in literature. Overall, the manuscript appears as a very immature first attempt to the scientific challenge of coupling division and genome segregation, and falls significantly short of the standards of a Nature Communication article.

In detail:

The authors claim high purity of ParM based on an SDS-PAGE shown in Fig. 2a. Despite the statement that only one band is shown, at least one additional band at approximately 13 kDa is visible. A quick densitometric analysis on the data with ImageJ of the bands at 39 and 13 kDa indicates that the band attributed to ParM accounts for less than 60 %. However, for an appropriate analysis of protein purity based on SDS-PAGE, it would be necessary to perform the measurement with a higher amount of protein (10 μ g). The supplemental figure of the elution shows even more additional bands. Also, the E. coli proteome comprises lots of proteins with higher masses than 97.4 kDa. It would be necessary to analyze the full lane of a PAGE able to display the entire cell lysate. Since remaining impurities after Ni-NTA are quite common, a second purification step with a SEC column is pretty much standard for protein purification in general but also for purifying ParM in particular. For purification, the authors reference a paper by Garner et al. (2007). However, this publication refers to an earlier publication by Garner in 2004 in the method section. This paper once again refers to van den Ent et al. (2002) which finally includes a full purification protocol. Surprisingly, van den Ent et al. use a completely different purification protocol than the authors: ammonium sulfate precipitation followed by gel filtration. This should be fixed. The ATPase assay is a good method to verify the protein function if the identity and purity of ParM is not confirmed by mass spectrometry. The authors are able to detect ATPase activity for their purified ParM. Unfortunately, they do not discuss their value; however, they reference a paper with

data on ATPase activity of ParM in the same paragraph. Jansen et al. (1997) observe an ATPase activity of 16.2 mol P_i per mol protein per hour which is more than 10 times higher than the value obtained by Zhao and Han. This difference should be discussed, since it suggests that the protein function of ParM used in this manuscript is below 10 %. Reduced protein function can be a result of impurities and aggregation. Notably, the authors do not differentiate between different actin-like proteins when referencing papers without indicating this in the manuscript. Those proteins come from different organisms and display huge differences in protein structure and filament architecture. Zhao and Han report higher ATPase activity of ParM upon addition of ParR and quote Forde et al (2017) who describe a system in which the ATPase activity of AlpC gets significantly enhanced by its adapter AlpA. In contrast, they also quote Jansen et al. (1997) who describe the ParMRC system from *E. coli* and report only slight stimulation by ParR alone. They show filaments formed by their protein and characterize them, but they should discuss the polymerisation behaviour of their ParM in greater detail since it differs from what was reported in literature. Garner et al. (2007) only observe long stabilized filaments in a complex with ParR and parC - without the other system components only short and transient seed filaments (~1.5 µm) are observed. This has been reproduced in many other publications/labs. The authors report very long and stable filaments even in the absence of ParR and parC, which seem to not largely affect filament sizes in their experiments. This is surprising, because they use a quite similar polymerization buffer with only two differences: no crowders and 10 times lower ATP concentrations. One would expect that both of those changes would lead to even smaller filaments. For Fig. 4c it would be better to choose a replicate with visible parC-beads, like in Fig. 4b, since yellow spots at the ends and sometimes the middle of the filaments are also detected in samples without beads (Suppl. Fig. 5 and 6). According to literature, the filaments stabilized by ParR and the DNA are longer because the complex acts as a cap and polymers attach to each other in an antiparallel manner. The protein in this manuscript seems to behave according to a different mechanism, therefore it is not possible to indirectly prove a bidirectional segregation event by polymer length. Also, labels which allow to distinguish between beads and lipids would be necessary. During image acquisition or processing, filaments were distorted (visible outside of GUVs, Fig. 3f).

Coupling the ParMRC system to PURE is a good approach to ensure functional offspring of dividing cells. A fluorescent protein (eGFP) is successfully produced by PURE after dividing GUVs. However, true evidence for an active role of ParMRC in the segregation of the eGFP gene is missing. Dreher et al. (2021) also receive two fluorescent daughter cells upon Ce6-mediated division without the use of any segregation system. And since large filaments are formed even in the absence of ParR and parC, it is not clear if and how many plasmids are captured and segregated by the filament in individual GUVs. The authors could prove their system's ability to actively deliver DNA to offspring by performing a statistical analysis on the daughter cell's fluorescence signals: GUVs with the ParMRC system should have a significantly higher amount of daughter cells with equal fluorescence intensities compared to vesicles without the protein system. Generally, data confirming the reproducibility of the system should be provided. How many GUVs of a population are able to form a single filament of desired length, contain the right amount of DNA and can separate two parC-sequences?

In Fig. 5b it is not clear why the eGFP DNA is expected to be visible at the ends of the filament. A biotin-modification of the DNA is mentioned but it is not explained why it exists and what it is supposed to bind to. Generally, the method part of the manuscript lacks important information. The design of ParM is not clear, since it is not noted which terminal end contains the His-tag, if the tag is cleavable or not and if there is a second modification for the fluorescence label. Cleavage of the tag, protein labelling, and coverslip preparation are entirely left out. Percentages of SDS-PAGES are not indicated.

Besides these severe technical shortcomings, the manuscript suffers from a very poor language quality. The immense number of grammar and spelling errors make it difficult to understand what Zhao and Han want to express. In the introduction it is repeatedly not clear if the literature is reviewed or something is claimed to be shown in the manuscript (e.g., l. 36-39). Some sentences are out of context and therefore not clear to the reader (l. 32-33). Moreover, important publications on ParMRC in the context of synthetic biology are not mentioned.

Generally, all references should be checked once again since their choice is not always clear. The authors reference articles which do not support their results without indicating this to the reader. Examples are already described for the purification and ATPase assay part. Apart from poorly discussing their results, they should make sure to quote original literature: As a reference for the complexity of eukaryotic mitotic spindles (l. 23-24) they reference a paper about actin-deformed artificial cells to mimic muscle contraction published by their own group (Li et al. 2022). Most importantly, the general narrative of the manuscript needs to be fixed, as the authors do not really mimic bacterial cell division. The GUV is split via Ce6-mediated division, i.e., achieved by photoinduced oxidative stress and osmotic shock, not based on any biology. Reactive oxygen produced by this is harmful to cell components (like DNA, proteins and lipids. Ce6-mediated division has previously been used by Dreher et al. (2021) as a chemical strategy to divide DNA encapsulating vesicles. Thus, the bacterial segregation machinery ParMRC discussed here does not support the division of the vesicle but rather acts as an obstacle during Ce6-mediated division, therefore being eliminated by laser irradiation prior to division. Therefore, the term 'bacterial cell division' is misleading.

Reviewer #4:

Remarks to the Author:

For the sake of clarity, the comments of the reviewer have been collated in black, and our response to each comment appears in blue. All the changes to the manuscript are highlighted in red.

Reviewer #1 (Remarks to the Author):

The manuscript by Han et al. demonstrates the reconstitution of a DNA segregation machinery into GUVs followed by GUV division. This is an impressive achievement. I see it as one of the most complex systems that have been reconstituted in bottom-up synthetic biology. The images speak for themselves, but in some parts the writing could be clearer. I have a few comments below, which should be addressed before publication.

Comments:

1. Storyline: Reading the beginning of the results section, I was worried that the paper is very superficial. But indeed, the data follows. It just has to be clear for the reader at the beginning of the results section that the first part describes the whole concept and the results will be presented later.

Thank the reviewer for the comments and valuable suggestion. We added below contents in page 4.

Figure 1 illustrates the mimicry of bacterial plasmid segregation and genetic information inheritance using an artificial cell that contains the ParMRC system and PURE system.The model genetic information of eGFP is inherited into daughter cells with the help of ParMRC system. The experimental data are discussed in the following results section.

2. Have the authors performed the controls for Ce6-mediated influx (addition of ATP without Ce6, addition of ATP and Ce6 without illumination)? ATP influx could also be confirmed by using fluorescent ATP or with a luciferase assay (no need to do this if the controls above are successfully performed).

Thank the reviewer for the comments and valuable suggestions. We followed the suggestion and carried out the control experiments. We added below figures (Supplementary Figure 14) and description in page 12.

In comparison, no ParM filaments inside the GUVs were observed in the absence of Ce6 upon the addition of 1 mM ATP and laser irradiation (Supplementary Figure 14 a) or in the presence of Ce6 and 1 mM ATP without laser irradiation (Supplementary Figure 14 b).

Supplementary Figure 14. The control experiments for Ce6-mediated ATP influx. **a** Fluorescence images of GUVs (labelled with NBD-PE) containing 9.6 μM ParM upon laser irradiation (405 nm, 0.3 mW, 5 s) at the presence of 1 mM ATP with no addition of Ce6. **a0**, **a1**, **a2**, and **a3** were taken before irradiation and 0 min, 5 min and 10 min after irradiation, respectively. **b** Fluorescence images of GUVs (labelled with NBD-PE) containing 9.6 μM ParM in the presence of Ce6 (10 μM) and 1 mM ATP with no laser irradiation. **b0**, **b1**, **b2**, and **b3** were taken before the addition of Ce6 and ATP, and 0 min, 5 min and 10 min after the addition of Ce6, respectively. The scale bars are 10 μm .

3. Is it really possible to split the ParM with 560nm? Have the authors tested this in bulk outside of the GUV? What is the mechanism? The authors link to Ref 4, but a bit more explanation in the manuscript text would be nice.

Thank the reviewer for the comments and valuable suggestion. We followed the suggestion and carried out experiments and added below contents in page 12.

Here, the ParM filaments were precisely cut into two parts both in solution (Supplementary Figure 15) and inside the GUV (Figure 3f) after strong laser irradiation (561 nm, 0.7 mW, 5 s), due to the laser-induced photochemical reaction of Cy3 dyes which were attached to ParM via chemical bonds. Upon laser irradiation, Cy3 dyes generate free radicals, which damage ParM proteins and consequently depolymerize ParM filaments⁴⁴.

Supplementary Figure 15. The split of a ParM filament in solution by laser irradiation. Fluorescence images of the ParM filament before (a) and after (b) laser irradiation (561 nm, 0.7 mW, 5 s). The scale bars are 5 μm .

4. It would also be nice to add in the discussion a paragraph about how ParM splitting could be achieved autonomously without laser irradiation.

Thank the reviewer for the comment and valuable suggestion. We added below sentences in page

12.

To divide the DNA in each pole of GUV into two daughter GUVs, it is essential to split the ParM filaments in the middle. In nature, *parC* DNA is pushed to opposite cell poles by ParM filaments with the help of ParR, followed by the detachment of ParR and *parC* DNA from ParM filaments; as a result, ParM-bound ATP underwent hydrolysis to depolymerize ParM filaments from their two ends¹⁸. However, the mechanism underlying ParR and *parC* DNA release from ParM filaments has not been investigated.

5. Figure 2 e, f: How was phosphate release and ATPase activity measured? What are the error bars? How often was this repeated?

Thank the reviewer for the comment and valuable suggestion. We added the description of the protocol to measure phosphate release and ATPase activity in page 26 as below. The error bars are shown in figure 2f. The experiments were repeated for 3 times, and the corresponding description was added in page 8 as below.

Page 26:

ParM ATPase activity assay. The ATPase assay of ParM was performed in accordance with the kit protocol. Under strong acid conditions, the phosphates released from ATP hydrolysed by ParM react with ammonium molybdate to produce phosphomolybdate yellows, which are reduced by stannous oxide (SnO) to phosphomolybdate blue. The intensity at 660 nm of the solution is proportional to the phosphate concentration. The ATPase activity of ParM was calculated using the following equation:

$$\text{ATPase activity of ParM} = \frac{C_{\text{standard}} * V_{\text{total}} * (A_{\text{experiment}} - A_{\text{control}})}{T * (A_{\text{standard}} - A_{\text{blank}}) * (C_{\text{ParM}} * V_{\text{sample}})}$$

where $A_{\text{experiment}}$, A_{control} , A_{standard} , and A_{blank} represent the absorbance at 660 nm of the experimental group, control group, standard sample and blank sample, respectively, with a C_{standard} of 0.5 $\mu\text{mol/mL}$, a V_{total} of 0.25 mL, a V_{sample} of 0.1 mL, and a T of 10 min.

Page 8:

e Phosphate release of ATP (4 mM) catalysed by different ParM concentrations (1.8 μM , 2.7 μM , 3.7 μM , 4.5 μM , 5.4 μM , 7.4 μM , 14.8 μM , 29.5 μM) within 1 hour. The phosphate release was obtained from three independent samples. The data are presented as the mean values \pm SDs; n=3. **f** The ATPase activity of ParM (6.7 μM) enhanced by ParR (7.7 μM , 23 μM , 38.4 μM , 76.8 μM , 115.1 μM); the pink, black, and blue columns represent the ATPase activity of ParM, ParR, and ParM-ParR, respectively. The ATPase activities of ParM, ParR, and ParM-ParR were obtained from three independent samples. The data are presented as the mean values \pm SDs (n = 3).

6. The authors study polymerization of ParM in droplets and GUVs (Figure 3c, d). Is it true that

for the droplets the ParM tends to localize at the periphery more? This is how it looks in the image. Do the authors have any idea why? Shouldn't that "consume" monomers such that the polymerization rate in droplets is expected to be slower compared to GUVs?

Thank the reviewer for the comments. According to our experimental results, the growth of ParM filament was random in the droplets. There is a set of experimental data (Figure R1) exhibiting no filament growth adjacent to the lipid monolayer.

Figure R1 Confocal microscopy images of ParM filament as a function of time inside the lipid protected droplet (green dotted circle for eye-guide) with ParM concentration of 9.6 μM and ATP concentration of 1 mM. The scale bars are 20 μm .

7. "In order to visualize parC DNA segregation under microscope, they were attached onto beads ($d=1352 \text{ nm}$)." How was the bead size measured so accurately? Use appropriate number of significant digits.

Thank the reviewer for the comment and valuable suggestion. The bead size was measured to be $1352 \pm 206 \text{ nm}$ by Zetasizer Nano (Malvern). We revised below sentence in page 14.

To visualize *parC* DNA segregation under a microscope, the *parC* DNAs were attached to beads ($d=1352 \pm 206 \text{ nm}$).

8. "The deformation rate of GUVs was $\sim 30 \%$ (Supplementary Figure14b). The division rate of deformed GUVs was $\sim 36 \%$ after laser irradiation." I assume the authors mean the fraction of deformed or divided GUVs. It does not make sense to talk about a rate here.

Thank the reviewer for the comment and valuable suggestion. We followed the suggestion and deleted this sentence.

Reviewer #2 (Remarks to the Author):

The authors describe the reconstitution of the bacterial ParMRC system in GUVs to drive localisation of plasmid DNA such that subsequent GUV division creates two daughter vesicles each containing the plasmid DNA. This represents minimal model artificial cell system for genetic transfer to daughter cells in artificial cell division.

ParM protein undergoes ATP dependent polymerisation into filamentous structures in the presence of the protein ParR that caps the filament end when bound to partner parC. ParC represents a DNA sequence within the target plasmid, thus anchoring the plasmid DNA to the poles of the filament. This assembly stabilises the growing filament with filament polymerisation thought to drive localisation of bacterial plasmid DNA to either end of a dividing cell, ensuring plasmid copies in both the resulting daughter cells. The ParMRC system is probably the most well characterised prokaryotic segregation machinery and function has also been widely demonstrated in vitro. The authors build on this underpinning work reconstituting the ParMRC system into GUVs and employing osmotic and chemical/light mediated mechanisms to drive vesicle breakup, creating an artificial cell system capable of DNA transfer into two daughter cells on forced division, analogous to the principles of the original bacterial system.

The authors demonstrate success by GFP plasmid localisation and expression by IVTT in both compartments following vesicle division. However, this reviewer feels additional comparative controls and further experimental details/characterisation are required before publication. However, the overall concept pushes beyond the current literature and the controlled molecular machinery mediated DNA segregation in artificial cells with division will be of broad interest and represents a significant achievement. This is however realised by combining two previously established protocols a) for in vitro reconstitution of ParMRC and b) controlled vesicle division, so one may argue the outcome is fairly obvious and a logical next step. In this context, the background literature in both the above areas should be better covered in the introduction to make the work's place in the field more clear. In addition, some scientific questions remain and the manuscript requires some extensive improvements to the text before the work is ready for publication, including clarification of methodology as it is often hard to interpret exactly what was done.

Major comments:

- 1. Appreciating English may not be the author's first language, there are places where the work is hard to interpret due to poor grammar or sentence construction combined with lack of detail. This should be improved throughout. It may be that some of the following questions are in fact addressed in the data, but this reviewer could not unpick some the experimental results and methodological descriptions to understand exactly what was done and what was being reported at times.*

Thank the reviewer for the comment and valuable suggestion. The English has been improved by professional language editing service “Nature publishing group language editing service”.

- A significant piece of prior work by the groups of Petra Schwille and Victor Sourjik should be acknowledged and described [ChemBioChem 2019, 20,2633–2642]. Whilst other works from these research groups are cited, this work doesn't appear to be. In this paper the researchers successfully reconstitute the ParMRC system with plasmid replication, using the T7 system to replicate circular plasmids with the Cre-lox system to regenerate them, and the ParMRC system for DNA segregation in vitro. ParMRC segregation is also demonstrated in biomimetic compartments driving plasmid localisation to the poles. Whilst these compartments are not engineered to undergo division, the underpinnings are highly relevant to the work under review and this represents important prior work that should be described and used to contextualise the advance being reported by the authors in the reviewed manuscript.*

Thank the reviewer for the comment and suggestion. We cited the work (ChemBioChem 2019, 20,2633–2642), and added the below sentence in page 2.

In addition to the *in vivo* study on the function of the ParMRC system, the *parC*-beads were shown to move in the opposite direction following ParM polymerization with the help of ParR *in vitro*¹⁹. The ParMRC partitioning system was successfully reconstituted into water-in-oil droplets to realize *in vitro* DNA segregation²⁰, which provided the possibility for plasmid segregation and inheritance in artificial cells.

- Vesicle division is an important element of the reported work, but only three or four lines are dedicated to this in the background. Whilst relevant literature is cited, there is no contextual*

detail for the reader. It would be appropriate to expand with some description of the mechanisms employed, how they work and results achieved – especially given the broad readership of nature communications who may not be familiar. The combination of vesicle dividing systems with DNA redistribution is only faintly touched upon “Some divisions were accompanied with DNA redistribution^{18,20,25,26}.” This should really be expanded upon to give a fair assessment of the current state of the art, briefly summarising what approaches have been employed, what results they have delivered and associated strengths or drawbacks. This would allow the reader to understand the place of the submitted work and the advantages of controlled localisation via ParMRC over other approaches.

Thank the reviewer for the comment and suggestions. We added below sentences in page 3.

The division of artificial cells is triggered by mechanical force²⁶, osmotic pressure^{27,28}, pH^{29,30} and active molecules³¹. The vesicles are symmetrically split by the sharp edge of a wedge-shaped splitter inside a microfluidic channel, which results from mechanical force²⁶. This method provided a strategy for symmetric, quick, efficient vesicle division. Phase separation of lipids in the membrane of artificial cells provides a division plane for dividing mother vesicles into two daughter vesicles through osmotic pressure in a controllable manner²⁸. Oleic acid in the inner leaflet of the GUVs (POPC/oleic acid) bilayer was deprotonated by the enzymatic reaction of urea-urease inside the vesicle, which enabled inner leaflet area variation and consequently led to division with the help of osmotic pressure³⁰. All the abovementioned vesicle divisions did not involve the natural division protein machinery. Bacterial division proteins including MinC, MinD, MinE, FtsA, and FtsZ were reconstituted into lipid vesicles³²⁻³⁴, attempting to divide artificial cells. Currently, the division of artificial cells is mostly designed for shape splitting, and some divisions are accompanied by DNA redistribution^{27,31,35,36}. GUVs that contain DNA grow and divide into daughter vesicles by synthesizing and incorporating phospholipids in situ upon the addition of vesicular membrane precursors³¹. DNA inside the ‘nucleus’ of eukaryote-like artificial cells is redistributed into two daughter artificial cells after division via osmotic stress²⁷. To date, researchers have found that all DNA redistribution in daughter artificial cells is random and passive and does not involve protein machinery or further translation of genetic information into DNA.

4. Detail on the Ce6 system should be added. This is fundamental to the study both in the import of ATP into the GUVs and it would seem the subsequent GUV division. However there is no explanation of what Ce6 is or how it works. “...upon light irradiation with the help of Ce6 molecules and osmotic pressure change.” is not a sufficient explanation for the mechanism of GUV division employed. Similarly “ATP molecules enter into GUVs through the transient pores due to the existence of Ce6.” Does not adequately describe to the reader the mechanism/role of Ce6 in the system for ATP influx into the GUV.

Thank the reviewer for the comments and suggestions. We added below sentences in page 10 and page 17.

Page 10:

Ce6 is a commonly used photosensitizer that is isolated from *Chlorella ellipsoidea*⁴¹. Reactive oxygen species are produced by Ce6 under laser (405 nm) irradiation (Supplementary Figure 8)^{35,42}, which causes unsaturated lipids to undergo peroxidation. This process affects membrane permeability because transient pores are formed in the lipid bilayer, allowing ATP to enter GUVs.

Page 17:

GUV division was achieved using the following two-step procedure: first, the GUVs were deformed into a dumbbell shape (Supplementary Figure 17) under hypertonic conditions, and second, Ce6-mediated peroxidation of unsaturated lipids was performed under laser (405 nm, 0.3 mW, 5 s) irradiation (Supplementary Figure 18)³⁵. Laser irradiation was focused on the central area of the dumbbell-shaped GUV, in which Ce6 molecules produced reactive oxygen species to oxidize local unsaturated lipids, resulting in a spontaneous increase in the membrane curvature and consequently splitting into two daughter GUVs.

- 5. How does the laser bring about splitting of the ParM filament? This is just described as “The ParM filaments was splited by laser (561 nm, 0.7 mW, 5 seconds) at center region of dumbbell GUV”. Is the laser illumination wide-field and the ParM just happens to split in the middle? Or is a method of laser scanning employed to specifically target the centre of the ParM filament? More methodological details are required to understand what was done and appreciate why the ParM filament splits. Has this previously been reported? Later it is described as “A strong laser irradiation (561 nm, 0.7 mW, 5s) precisely cut ParM filament into two parts inside GUVs (Figure 3f), because the laser induced photochemical reaction of cy3 modified onto ParM.” This is not clear and there is no insight provided into the mechanism. It is not obvious how ref 4 relates to this and it should be noted that biomolecules tagged with the fluorophore cy3, which is excited at 561nm are often exposed to powers far in excess of 0.7 mW, e.g. in single molecule imaging, without apparent reaction beyond the expected fluorescence processes. Does this higher laser power of 0.7 mW pose any problems with driving the Par system (or the beads?) to the GUV membrane given the manuscript states that laser powers above 0.3 mW were seen to to drive association of Par to the membrane (see later comment on this also).*

Thank the reviewer for the comments. A method of laser scanning was employed to specifically target the center of the ParM filaments to split them. The split of ParM filaments by laser irradiation was reported in the literature (Science 2007 315, 1270-1274), which was cited as reference 19. In single molecule imaging using the fluorophore Cy3, although the samples were exposed to more powerful laser, the exposure time (e.g. 20 ms (Nature microbiology 2021, 6, 553)) was far more less. Therefore, no apparent reaction happened beyond the expected fluorescence processes. While the total energy for ParM filament splitting was much higher due to the longer exposure time.

When we stated that laser powers above 0.3 mW were seen to drive association of Par to the membrane, the laser was 405 nm (rather than 561 nm) to induce Ce6 molecules (rather than cy3) to peroxidase lipid to form transient pores in the membrane to allowing ATP inflow, causing the high ATP concentration adjacent to the GUV membrane, further resulting the ParM filaments to grow adjacent to the inner bilayer membrane. In addition, we applied the laser (561 nm, ~ 1 μ m in

diameter) in the center area to split ParM filament inside GUVs (~ 20 μm in diameter). Therefore, we did not observe the problems caused by the laser (561 nm) irradiations with driving the ParMRC system to the GUV membrane. The splitting mechanism was explained as below in page 12 and 24.

Page 12:

Here, the ParM filaments were precisely cut into two parts both in solution (Supplementary Figure 15) and inside the GUV (Figure 3f) after strong laser irradiation (561 nm, 0.7 mW, 5 s), due to the laser-induced photochemical reaction of Cy3 dyes which were attached to ParM via chemical bonds. Upon laser irradiation, Cy3 dyes generate free radicals, which damage ParM proteins and consequently depolymerize ParM filaments⁴⁴.

Supplementary Figure 15. The split of a ParM filament in solution by laser irradiation. Fluorescence images of the ParM filament before (a) and after (b) laser irradiation (561 nm, 0.7 mW, 5 s). The scale bars are 5 μm .

Page 24:

Splitting of ParM filaments by laser irradiation. ParM filaments were precisely cut into two parts both in solution and inside the GUVs, using high-intensity laser irradiation (561 nm, 0.7 mW, 5 s) through laser scanning at the central area (~ 1 μm in diameter) of the filaments several times until full splitting was achieved. The laser intensity was measured by an optical densitometer (CEL-NP2000).

6. *The most important experimental questions and comments concern the final figure and the expression of eGFP following ParMRC segregation. The production of parC-eGFP coated beads is not described. Presumably this is the same as the earlier parC beads? This is essential to know given the reliance on the eGFP expression data to evidence the main claim of the manuscript, that of controlled DNA segregation by the ParMRC system prior to artificial cell division and subsequent gene expression. The reader has to assume what was done, which is not desirable for any presented data, especially not the key result. In this context there are several important questions that are not obviously answered by the controls and data/text presented concerning the eGFP expression experiments.*

1) *Is there any chance of free eGFP plasmid or unbound parC-eGFP coated beads being present inside the GUV and ending up in the two daughters simply by diffusion? Have control experiments without the ParM polymerisation been conducted to see what proportion of daughter vesicles are able to express eGFP after division with no active segregation? If this is possible, then how do we know for certain the presented result is (entirely) due to the ParMRC system?*

Comparing to the outcome with and without ParM(RC) would confirm the role of the Par system, where one would expect to observe an increase in probability of both daughters expressing eGFP with the ParMRC system. This comparison would give confidence in confirming the necessity of ParMRC to achieve the reported result of expression in both daughters. It would also allow quantitative comparison of the efficiency increase of the process due to ParMRC over random chance. To assist with understanding here the authors could estimate how many parC-eGFP coated beads they expect per vesicle. Will these likely all be bound to ParM filaments, or would some be expected to be free? Would imaging reveal the bound efficiency here? The number of anticipated plasmids per bead could also be estimated and correlated to expression rate. The presence or absence of free parC-eGFP plasmid could also easily be confirmed experimentally to rule out equilibration of free plasmid in the vesicle prior to division.

Thank the reviewer for the comments and valuable suggestions. We addressed each comment separately as below for clarity.

(1) *Is there any chance of free eGFP plasmid or unbound parC-eGFP coated beads being present inside the GUV and ending up in the two daughters simply by diffusion?*

The excess unbonded *parC* DNAs were fully removed by magnetic separation method, which was confirmed by both UV-vis spectrometer and Agarose gel electrophoresis (Supplementary Figure 28). Therefore, no free eGFP plasmids were encapsulated inside GUVs. We added below sentence in page 28. The calculation of the amount of eGFP plasmid per bead refers to following question (4).

Since the amount of eGFP plasmid on each bead (~ 0.87 ng) is sufficient for expression in GUVs containing the PURE system, we encapsulated two *parC*-beads in each GUV as much as possible; as a result, the beads bound to the two ends of ParM filaments, avoiding the free unbonded beads in the GUV.

(2) *Have control experiments without the ParM polymerisation been conducted to see what proportion of daughter vesicles are able to express eGFP after division with no active segregation? If this is possible, then how do we know for certain the presented result is (entirely) due to the ParMRC system? Comparing to the outcome with and without ParM(RC) would confirm the role of the Par system, where one would expect to observe an increase in probability of both daughters expressing eGFP with the ParMRC system. This comparison would give confidence in confirming the necessity of ParMRC to achieve the reported result of expression in both daughters. It would also allow quantitative comparison of the efficiency increase of the process due to ParMRC over random chance.*

We followed the suggestion and carried out control experiments for eGFP expression in daughter GUVs with no active segregation (ParMRC system), and added corresponding description as below in page 19.

In comparison, no green fluorescence was observed at 37 °C for 3 h inside daughter cells when the mother cells did not contain the PURE system (Supplementary Figure 24). Without the ParMRC system, the *parC*-eGFP DNA beads were randomly distributed into two daughter GUVs

(Supplementary Figure 25), which resulted in a significant difference in the expression rate between the two daughter GUVs (Supplementary Figure 26). In contrast, the expression rates of each daughter GUV with the ParMRC system were almost identical (Supplementary Figure 26), which further confirmed the role of the ParMRC system in segregating genetic information in both daughter GUVs.

Supplementary Figure 25. Confocal microscopy images of GUV division and follow-on eGFP expression in two daughter GUVs without ParMRC system. Time-dependent fluorescence images of GUV division at 0 s (a), 43 s (b), 86 s (c), and eGFP expression at 37 °C in daughter cells at 90 min (d), 180 min (e), 210 min (f), respectively. The scale bars are 10 μm.

Supplementary Figure 26. The normalized fluorescence intensity of eGFP in daughter GUV 1 and GUV 2 with or without ParMRC system. The control experiments represent the condition of eGFP expression in daughter GUV 1 and GUV 2 without the ParMRC system. The normalized fluorescence intensities of eGFP are from three independent samples. Data are presented as the mean values \pm SDs, $n=3$. Statistical analyses were carried out by unpaired two-tailed student's t-test. $**p < 0.01$. Source data are provided as a Source Data file. The ParMRC experiments represent the condition of eGFP expression in daughter GUV 1 and GUV 2 with the ParMRC system. The normalized fluorescence intensities of eGFP are from three independent samples. Data are presented as the mean values \pm SDs, $n=3$. Statistical analyses were carried out by unpaired two-tailed student's t-test. NS, not significant ($P = 0.9670$).

(3) *To assist with understanding here the authors could estimate how many parC-eGFP coated beads they expect per vesicle. Will these likely all be bound to ParM filaments, or would some be*

expected to be free? Would imaging reveal the bound efficiency here?

We followed the suggestion added below contents in page 28.

The GUVs were collected at the bottom of the tube and used for subsequent experiments. ParMRC-containing GUVs were prepared through a similar method as that used for the ParM-containing GUVs. Since the amount of eGFP plasmid on each bead (~ 0.87 ng) is sufficient for expression in GUVs containing the PURE system, we encapsulated two *parC*-beads in each GUV as much as possible; as a result, the beads bound to the two ends of ParM filaments, avoiding the free unbonded beads in the GUV.

(4) The number of anticipated plasmids per bead could also be estimated and correlated to expression rate. The presence or absence of free parC-eGFP plasmid could also easily be confirmed experimentally to rule out equilibration of free plasmid in the vesicle prior to division.

We estimated the number of plasmids on each bead and added below contents in page 18 and page 26.

Page 26:

The number of plasmids on each bead was estimated using the following calculations. The concentration of streptavidin beads was $(1.653 \pm 0.006) \times 10^6/\text{mL}$ using an automated cell counter. The concentration of biotinylated *parC* DNA was $466 \pm 24 \text{ ng}/\mu\text{L}$, as determined via UV-vis spectroscopy. The mixture of streptavidin beads and biotinylated *parC* DNA was incubated for 30 min at room temperature. After magnetic separation, the supernatant was removed and the *parC* DNA concentration was determined to be $419 \pm 1 \text{ ng}/\mu\text{L}$. The *parC*-beads were washed again using the same protocol. The *parC* DNA concentration in the supernatant was $1.17 \pm 0.29 \text{ ng}/\mu\text{L}$. After the third wash, the concentration of *parC* DNA in the supernatant was 0. Thus, the total amount of plasmid bound to the beads was calculated to be 72191.5 ng by subtracting the remaining plasmid DNA in the supernatants ($419 \text{ ng}/\mu\text{L} * 50 \mu\text{L} + 1.17 \text{ ng}/\mu\text{L} * 50 \mu\text{L}$) from the added plasmid DNA ($466 \text{ ng}/\mu\text{L} * 200 \mu\text{L}$). The average amount of *parC* DNA was ~ 0.87 ng when 72191.5 ng was divided over 82650. This amount of DNA single beads was sufficient to trigger eGFP expression using PURE systems.

Page 18:

The full mimicry of bacterial plasmid segregation and inheritance was completed using artificial cells that contained the ParMRC system and PURE system (Figure 5c). The biotinylated *parC*-eGFP was prepared using a method similar to that used for biotinylated *parC*. The amount of biotinylated *parC*-eGFP DNA on each bead was estimated to be ~ 0.87 ng, which was enough to trigger eGFP expression using PURE systems.

2) In Fig5b are the contents of the two vesicles definitely separated given the two vesicles are still in contact for at least 90 mins at the elevated temperature of 37 degrees used to initiate IVTT

protein expression. Given the presence of Ce6 and its use to make pores in the membrane earlier in the protocol could plasmid or eGFP be exchanged via pores? Similarly in images B6 and B7 it looks the two vesicles are linked by a membrane tether. Could the two vesicles shown still be in diffusive contact? How repeatable was this result as only one example is shown in the manuscript. Was this experiment successful more than once? In the example shown, in b3 is there a large third vesicle in the top left had corner of the image that the original vesicle(s) seem attached to? What is happening here?

Thank the reviewer for the comments and valuable suggestions. We followed the suggestion and carried out experiments to verify no penetration of the *parC*-beads and eGFP via the transient pores in the presence of Ce6 upon laser irradiation, and added corresponding description as below in page 15 and page 18. The experiments of the expression of eGFP DNA in daughter GUVs with help of the ParMRC system were successfully repeated three times. We have replaced the previous Figure 5b with the new set of data as below. The GUV at the bottom-right corner of Figure 5b1 and b2 just appeared when taking image, which moved away since it disappeared on the following images. The statistical analysis of the expression of eGFP in daughter GUVs with ParMRC system was shown in Supplementary Figure 26.

Page 15:

The *parC* DNA inside the GUV did not leak through the transient pores formed by Ce6 and laser irradiation (Supplementary Figure 16). Thus, DNA segregation was successfully mimicked, which paved the way for mimicking the inheritance of genetic material after bacterial division.

Supplementary Figure 16. Confocal microscopy images of the GUV containing *parC*-bead. Time-dependent fluorescence images of the GUV containing the *parC*-bead after the addition of Ce6 (10 μM) and laser irradiation (405 nm, 0.3 mW, 5 s) at 0 min, 60 min, 120 min, 180 min, 300 min, respectively. The scale bars are 10 μm.

Page 18:

The expression of eGFP in solution was further confirmed by fluorescence spectrometry (Supplementary Figure 21b). eGFP was also expressed inside the GUV since the GUV turned green gradually within ~ 4 hours at 37 °C (Supplementary Figure 22). The eGFP inside the GUV did not leak through the transient pores formed by Ce6 and laser irradiation (Supplementary Figure 23).

Supplementary Figure 23. Confocal microscopy images of the GUV containing eGFP. Time-dependent fluorescence images of eGFP inside the GUV after the addition of Ce6 (10 μM) and laser irradiation (405 nm, 0.3 mW, 5 s) at 0 min, 60 min, 120 min, 180 min, 240 min, 300 min, respectively. The scale bars are 10 μm .

Page 19:

Figure 5 b Confocal microscopy images of GUV division (**b1** to **b3**) and eGFP expression at 37 $^{\circ}\text{C}$ (**b4** to **b7**). **b1**, **b2** and **b3** indicate GUV deformation, filament splitting, and division into two daughter cells, respectively. The white arrows in **b1** and **b2** indicate *parC*-eGFP DNA. The ParM filament was split by laser irradiation (561 nm, 0.7 mW, 5 s). The scale bars are 10 μm . **c** Schematic illustration of eGFP expression in two daughter cells. The corresponding fluorescence intensity in daughter GUV 1 (**d**) and daughter GUV 2 (**e**) as a function of time. **The normalized fluorescence intensity of eGFP was calculated from three independent samples. The data are presented as the mean values \pm SDs; $n=3$.**

3) line 264 “no green fluorescence was observed at 37 $^{\circ}\text{C}$ for 3 h inside daughter cells when the mother cell did contain PURE system (Supplementary Figure20).” - presumably this should say did not?

Thank the reviewer for the comments and valuable suggestion. We corrected the sentence as below in page 19. Previous Supplementary Figure20 became Supplementary Figure 24 in revised version.

In comparison, no green fluorescence was observed at 37 °C for 3 h inside daughter cells when the mother cells did **not** contain the PURE system (**Supplementary Figure 24**).

Minor comments:

- 1. The nature of the ParMRC system could be better described for the non-expert, as understanding of this is essential for understanding the reported experiments.**

Thank the reviewer for the comments and valuable suggestions. We added corresponding description as below in page 2.

Cell division is a key feature of life¹, and it is crucial that daughter cells inherit accurate and correct genomic material to maintain the species². Eukaryotes possess complicated chromosome segregation machinery in mitotic spindles³⁻⁵. In prokaryotes, relatively simple mechanisms are used to partition newly replicated DNA⁶⁻⁹. The ParMRC partitioning system is among the most representative systems for bacterial DNA segregation^{6,10-12}. This system consists of three elements, namely, ParM (an actin-like protein), ParR (an adapter protein) and *parC* (a centromere-like site)¹³⁻¹⁵. ParM is an ATPase that can bind ParR, while ParR can specifically bind to *parC*. The genomic materials (plasmids) are segregated into opposite poles of cells by the polymerization of ParM with the help of ParR and *parC* in the presence of ATP¹⁶. **This process is important because it ensures that genetic material is evenly distributed among daughter cells prior to division. In a previous study, bipolar elongation of ParM filaments was observed in a bacterial cell through immunofluorescence microscopy¹⁷. Through time-lapse fluorescence microscopy, researchers observed dynamic segregation of plasmids towards the poles of cells by ParM polymerization, which was facilitated by ParR¹⁸. Once plasmids reach the poles of the cell, the ParM filaments disassemble spontaneously and distributes plasmids to each daughter cell after cell division. In addition to the in vivo study on the function of the ParMRC system, the *parC* beads were shown to move in the opposite direction following ParM polymerization with the help of ParR in vitro¹⁹. The ParMRC partitioning system was successfully reconstituted into water-in-oil droplets to realize in vitro DNA segregation²⁰, which provided the possibility for plasmid segregation and inheritance in artificial cells.**

- 2. The grammar in the introduction makes it not always clear whether the work of the paper or the work of others is being described. This can be worked deduced, but improved construction would help the reader.**

Thank the reviewer for the comments. The English has been improved by professional language editing service “Nature publishing group language editing service”.

3. *Figure 1 is described and presented as if it were an experimental result, but is just a cartoon schematic. Discussing as a result at this stage of the manuscript makes the flow feel muddled when the subsequent data is gel purification. A schematic is a good idea for figure one but the text around it could be improved to help the reader and the flow of the manuscript.*

Thank the reviewer for the comments and valuable suggestions. We added below contents in page 4.

Figure 1 illustrates the mimicry of bacterial plasmid segregation and genetic information inheritance using an artificial cell that contains the ParMRC system and PURE system.The model genetic information of eGFP **is** inherited into daughter cells with the help of ParMRC system. **The experimental data are discussed in the following results section.**

4. *Figure 2f - what is the 'control' condition of the black bars. Not clear from the text or legend label of just control.*

Thank the reviewer for the comments. The 'control' condition of black bars represented the ATPase activity of ParR. We added below contents in page 9.

f The ATPase activity of ParM (6.7 μM) enhanced by ParR (7.7 μM , 23 μM , 38.4 μM , 76.8 μM , 115.1 μM); **the pink, black, and blue columns represent the ATPase activity of ParM, ParR, and ParM-ParR, respectively. The ATPase activities of ParM, ParR, and ParM-ParR were obtained from three independent samples. The data are presented as the mean values \pm SDs (n = 3).**

5. *In characterising the polymerisation behavior of ParM, why not represent all measured lengths in a histogram rather than categorising into arbitrary categories of different size ranges?*

Thank the reviewer for the comments and valuable suggestion. We followed the suggestion and revised Figure 3b as below. Box-and-whisker plots were used according to the journal editorial

policy. The corresponding description was added in page 10.

The length distributions of ParM filaments at different ParM concentrations (2.4 μM , 4.8 μM , 9.6 μM , 14.3 μM , and 19.1 μM) were investigated. With increasing ParM concentration, the filament length became longer (Figure 3b). The average lengths of the filaments were $2.9 \pm 1.2 \mu\text{m}$, $5.3 \pm 1.8 \mu\text{m}$, $8.3 \pm 3.3 \mu\text{m}$, $15.2 \pm 7.2 \mu\text{m}$ and $42.9 \pm 25.4 \mu\text{m}$ at ParM concentrations of 2.4 μM , 4.8 μM , 9.6 μM , 14.3 μM and 19.1 μM , respectively (Figure 3b, Supplementary Figure 6). The average filament length of $8.3 \pm 3.3 \mu\text{m}$ at a ParM concentration of 9.6 μM fit the GUV size well (from 10~20 μm in diameter; Supplementary Figure 7). Therefore, 9.6 μM ParM was chosen for segregating DNA molecules inside the GUV.

Figure 3. b ParM filament length at ParM concentrations of 2.4 μM , 4.8 μM , 9.6 μM , 14.3 μM , and 19.1 μM . The concentration of ATP was 1 mM. The lengths of ParM were determined from 105 independent samples at each concentration. $n = 105$. The central line corresponds to the median. The black dots correspond to the mean values. The lower and upper hinges of the boxes correspond to the 25th and 75th percentiles, respectively, and the whiskers represent the $1.5 \times$ interquartile range extending from the hinges. Statistical analyses were carried out by an unpaired two-tailed Student's t test. *****) $P < 0.00001$.

6. The text states “Ce6 incorporated spontaneously into the outer leaflet of GUV membrane (Figure S8)” It is not possible to tell from a microscopy image (Figure S8) of a vesicle whether the Ce6 has incorporated in the outer, inner or both leaflets. The membrane thickness is far smaller than the diffraction limit and the increased observed annulus thickness of the GUV is most likely a focus effect or optical artefact.

Thank the reviewer for the comments. We fully agree with the reviewer and deleted the sentence of “Ce6 incorporated spontaneously into the outer leaflet of GUV membrane (Supplementary Figure 8)” and Supplementary Figure 8 in previous version.

7. *The text states “In order to determine the Ce6 concentration and light strength of laser, the 8- Hydroxypyrene-1,3,6-trisulfonic acid trisodium salt (HPTS) diffusion coefficient (D) across the lipid bilayer was investigated as a function of Ce6 concentration and laser intensity.” This measurement does not report on laser power or Ce6 concentration, but on the effect of these two variables on the membrane.*

Thank the reviewer for the comments and valuable suggestions. We revised below sentences in page 10.

To determine the effect of Ce6 concentration and light strength of the laser on membrane permeability, the 8-hydroxypyrene-1,3,6-trisulfonic acid trisodium salt (HPTS) diffusion coefficient (D) across the lipid bilayer was investigated as a function of Ce6 concentration and laser intensity.

8. *In the same section “Polymerization behavior of ParM” In the discussion of the rationale behind FigS12 and S13, the authors are showing and discussing a range of experiments which were designed to inform the most appropriate laser power at which to carry into the next phase of the work - e.g. HPTS diffusion at varying laser powers is used as a measure of extent of pore formation, why not show the experimental results on which this decision on laser power was actually made as it is stated “ParM was found to polymerize adjacent to the lipid bilayer of GUVs with the laser power over 0.3 mW”? Which means most of the tested laser powers were ruled out. A lot of text in the main manuscript and data in the SI is dedicated characterising the relationship between laser power and HPTS diffusion, but no data is given to support the decision that was actually made.*

Thank the reviewer for the comments and valuable suggestions. We followed the suggestion and carried out experiments of the influence of different laser intensity on the polymerization of ParM inside the GUV. We added new Supplementary Figure 13 and modified below sentences in page 11.

A faster penetration rate was observed with higher laser powers. ParM was found to polymerize adjacent to the lipid bilayer of GUVs with a laser power greater than 0.3 mW (Supplementary Figure 13). Therefore, a laser power of 0.3 mW was chosen for the polymerization of ParM in the GUV system to segregate DNA.

Supplementary Figure 13. The polymerization of ParM inside GUVs at different laser intensities. **a** Schematic illustration of ParM polymerization inside GUVs upon the laser irradiation with power over 0.3 mW. Confocal microscopy images of ParM polymerization adjacent to the lipid bilayer of GUVs as a function of time upon 5 s laser (405 nm) irradiation with the power of 0.5 mW (**b**), 0.7 mW (**c**), and 0.9 mW (**d**), respectively. ParM concentration inside GUVs and ATP concentration outside GUVs are 9.6 μM and 1mM respectively. The scale bars are 10 μm .

9. *Figure 3. It would be helpful to label c) as droplet and d) as vesicles. The text discussing the results of Fig3 states that filaments inside GUVs were smaller than in lipid coated droplets- can this really be confidently be concluded from the data of Fig3e with such large variance and just three replicates (n=3)?*

Thank the reviewer for the comments. We followed the formatting guide of Nature Communications to label figures with bold number. We fully agree with the reviewer, and deleted comparison description of “which is smaller than that in the droplets.”

10. *Can the bead diameter really be measured as accurately as $d=1352\text{ nm}$?*

Thank the reviewer for the comments and valuable suggestions. The size of beads was measured to be $1352 \pm 206\text{ nm}$ by Zetasizer Nano (Malvern). We revised below sentence in page 14.

To visualize *parC* DNA segregation under a microscope, the *parC* DNAs were attached to beads

($d=1352 \pm 206$ nm).

11. Throughout it is not always totally clear what we are looking at in the microscopy images as the descriptions are not complete. In FigS14, what constituted deformation in the assessment employed?

Thank the reviewer for the comments and valuable suggestions. We added blew contents in the figure caption of previous Fig S14 (Supplementary Figure 17 in revised manuscript).

Supplementary Figure 17. Confocal microscopy images of deformed GUV. The deformation of GUVs was triggered by the osmotic pressure caused by 70 mM difference between outside (330 mM glucose) and inside GUVs (260 mM sucrose). GUVs membranes were labeled with NBD-PE (green fluorescence). The GUVs in the red rectangles are deformed GUVs (dumbbell shape GUVs). The scale bar is 10 μm.

12. In. FigS20. The figure is titled “Confocal microscopy images of eGFP expression in two daughter cells without PURE system.” this should be revised as there is no expression system and so no expression.

Thank the reviewer for the comments and valuable suggestion. We revised below contents figure caption of previous Supplementary Figure 20 (Supplementary Figure 24 in revised manuscript).

Supplementary Figure 24. Confocal microscopy images of two daughter cells without PURE system.

13. There are further examples of unclear statements in the main text and figure captions where clarification would show off the authors work a lot better and save the reader having to wrestle so much with the information to really understand the work.

Thank the reviewer for the comments. The English has been improved by professional language editing service “Nature publishing group language editing service” to improve the clarity of the manuscript.

This document certifies that the manuscript

Artificial cells containing Par system for bacterial plasmid segregation and inheritance mimicry

prepared by the authors

Jingjing Zhao, Xiaojun Han*

was edited for proper English language, grammar, punctuation, spelling, and overall style by one or more of the highly qualified native English speaking editors at SNAS.

This certificate was issued on **January 13, 2024** and may be verified on the SNAS website using the verification code **D8FC-B3F8-2BA1-49CC-7480**.

Neither the research content nor the authors' intentions were altered in any way during the editing process. Documents receiving this certification should be English-ready for publication; however, the author has the ability to accept or reject our suggestions and changes. To verify the final SNAS edited version, please visit our verification page at secure.authorservices.springernature.com/certificate/verify. If you have any questions or concerns about this edited document, please contact SNAS at support@as.springernature.com.

Reviewer #3 (Remarks to the Author):

In their generally intriguing attempt towards mimicking synthetic cell division, Zhao and Han aim to combine the bacterial ParMRC segregation system with the PURE system to couple controlled genetic inheritance with vesicle splitting and protein expression in daughter vesicles. The objective to segregate DNA in a controlled way into dividing vesicles is timely and promising. However, the execution is unsatisfying, as the authors do not provide evidence for a significant role of the ParMRC system in the distribution of the DNA (eGFP gene) to offspring cells, leave alone a functional coupling to the (forced) division, as would be a truly noteworthy accomplishment. Furthermore, there are major deficiencies in the quality of the produced ParM and discrepancies between its behaviour shown here and in literature. Overall, the manuscript appears as a very immature first attempt to the scientific challenge of coupling division and genome segregation, and falls significantly short of the standards of a Nature Communication article.

In detail:

- 1. The authors claim high purity of ParM based on an SDS-PAGE shown in Fig. 2a. Despite the statement that only one band is shown, at least one additional band at approximately 13 kDa is visible. A quick densitometric analysis on the data with ImageJ of the bands at 39 and 13 kDa indicates that the band attributed to ParM accounts for less than 60 %. However, for an appropriate analysis of protein purity based on SDS-PAGE, it would be necessary to perform the measurement with a higher amount of protein (10 µg). The supplemental figure of the elution shows even more additional bands. Also, the E. coli proteome comprises lots of proteins with higher masses than 97.4 kDa. It would be necessary to analyze the full lane of a PAGE able to display the entire cell lysate. Since remaining impurities after Ni-NTA are quite common, a second purification step with a SEC column is pretty much standard for protein purification in general but also for purifying ParM in particular. For purification, the authors reference a paper by Garner et al. (2007). However, this publication refers to an earlier publication by Garner in 2004 in the method section. This paper once again refers to van den Ent et al. (2002) which finally includes a full purification protocol. Surprisingly, van den Ent et al. use a completely different purification protocol than the authors: ammonium sulfate precipitation followed by gel filtration. This should be fixed.***

Thank the reviewer for the comments and valuable suggestions. We followed the suggestion to purify ParM with SEC column after Ni-NTA column, and to run SDS PAGE from 250 kDa to 10 kDa. We replaced the previous Figure 2a and b with new SDS PAGE data of ParM and ParR in page 8 and added corresponding description as below in page 23. We also fixed the issue of purification protocol citation by citing the correct one as below in page 6.

Page 8:

Figure 2. SDS/polyacrylamide gel images of ParM (39 kDa) (a) and ParR (14 kDa) (b).

Page 23:

Expression and Purification of ParM and ParR. The ParM gene was synthesized and cloned and inserted into the expression vector pET28a (a plasmid encoding an N-terminal 6-histidine tag) to generate the ParM plasmid (supplementary methods 1), which was subsequently transformed into *E. coli* BL21 (DE3) cells at a certain concentration (~10 ng). ...The ParM was purified by eluting with a series of buffer solutions containing different concentrations (50 mM, 100 mM, 200 mM, 300 mM, 400 mM, and 500 mM) of imidazole. The ParM in different fractions was analysed via SDS-PAGE. The relatively pure ParM fraction (200 mM imidazole) was further purified by a size-exclusion chromatographic column in 260 mM sucrose, 30 mM Tris-HCl, 2 mM MgCl₂, 1 mM DTT, and 100 mM KCl at pH 7.5. The obtained ParM was concentrated and stored at -80 °C. The ParR gene was synthesized and cloned and inserted into the expression vector pET28a to obtain the ParR plasmid (supplementary methods 2). ParR was extracted using a method similar to that used for ParM. The purities of ParM and ParR were greater than 99%, as determined by densitometric analysis with ImageJ. The N-terminal 6-histidine tag was not cleaved before use. The extracted proteins were stored at -80 °C.

Page 6:

ParM and ParR were overexpressed in *Escherichia coli* BL21 (DE3) cells and purified using previously described protocols^{37,38}.

2. *The ATPase assay is a good method to verify the protein function if the identity and purity of ParM is not confirmed by mass spectrometry. The authors are able to detect ATPase activity for their purified ParM. Unfortunately, they do not discuss their value; however, they reference a paper with data on ATPase activity of ParM in the same paragraph. Jansen et al. (1997) observe an ATPase activity of 16.2 mol Pi per mol protein per hour which is more than 10 times higher than the value obtained by Zhao and Han. This difference should be discussed, since it suggests that the protein function of ParM used in this manuscript is below 10 %. Reduced protein function can be a result of impurities and aggregation. Notably, the authors do not differentiate between different actin-like proteins when referencing papers without indicating this in the manuscript. Those proteins come from different organisms and display huge differences in protein structure and filament architecture. Zhao and Han report higher ATPase activity of ParM upon addition of ParR and quote Forde et al (2017) who*

describe a system in which the ATPase activity of AlpC gets significantly enhanced by its adapter AlpA. In contrast, they also quote Jansen et al. (1997) who describe the ParMRC system from E. coli and report only slight stimulation by ParR alone.

Thank the reviewer for the comments and valuable suggestions. We measured the ATPase activity of ParM purified by Ni-NTA and SEC column to be $16.58 \pm 0.01 \mu\text{M}$ per μM (ParM) per hour, which was similar to the previously reported (Journal of molecular biology 1997, 269, 505-513). In our paper, the ParM protein was from *E. coli*. Therefore, we deleted the reference (Forde et al (2017)) containing ParM protein from other species. With the addition of $2 \mu\text{g}$ ParR, the ATPase activity of ParM was enhanced 1.64-fold in our paper, whilst the ATPase activity of ParM was enhanced ~ 1.3 fold. We added below sentences in page 7 as below.

The rate of ATP hydrolysis was estimated to be $16.58 \pm 0.01 \mu\text{M}$ per μM (ParM) per hour, **which was similar to the previously reported value of $16.2 \mu\text{M}$ per μM (ParM) per hour⁴⁰.**

- 3. They show filaments formed by their protein and characterize them, but they should discuss the polymerisation behaviour of their ParM in greater detail since it differs from what was reported in literature. Garner et al. (2007) only observe long stabilized filaments in a complex with ParR and parC - without the other system components only short and transient seed filaments ($\sim 1.5 \mu\text{m}$) are observed. This has been reproduced in many other publications/labs. The authors report very long and stable filaments even in the absence of ParR and parC, which seem to not largely affect filament sizes in their experiments. This is surprising, because they use a quite similar polymerization buffer with only two differences: no crowders and 10 times lower ATP concentrations. One would expect that both of those changes would lead to even smaller filaments.*

Thank the reviewer for the comments and valuable suggestions. We followed the suggestion and carried out experiments to investigate the polymerization behavior of ParM with or without crowder at different ATP concentrations. The long ParM filaments were also obtained in the absence of ParR and *parC* in the literatures (Biochemical and biophysical research communications 2007, 353, 109-114. EMBO Journal 2008, 27, 570-579). We added below sentences in page 10 as below.

The threshold ATP concentration for the formation of ParM filaments varied with ParM concentration (Figure 3a), in which the green dots indicate the successful formation of ParM filaments under these conditions and the cross symbols indicate the failure of ParM filament formation. **The influence of the crowder on ParM polymerization was also investigated. The ParM filaments were shorter after treatment with 10 mM ATP, regardless of the presence of the crowder (0.4% methylcellulose) (Supplementary Figure 5a and b). ParM filaments were thicker in the presence of methylcellulose (0.4%) (Supplementary Figure 5c) at 1 mM ATP than in the absence of methylcellulose (0.4%) at 1 mM ATP (Supplementary Figure 5d). In the following experiments, no crowder was used for the ParM polymerization.**

Supplementary Figure 5. The ParM polymerization behavior under different conditions. Fluorescence microscopy images of ParM (9.6 μM) filaments polymerized in the buffer solution (10 mM ATP, 30 mM Tris-HCl, 2 mM MgCl_2 , 1 mM DTT, and 100 mM KCl at pH 7.5) containing 0.4% methylcellulose (a) and 0 % methylcellulose (b). Fluorescence microscopy images of ParM (9.6 μM) filaments polymerized in the buffer solution (1 mM ATP, 30 mM Tris-HCl, 2 mM MgCl_2 , 1 mM DTT, and 100 mM KCl at pH 7.5) containing 0.4% methylcellulose (c) and 0 % methylcellulose (d). The scale bars are 20 μm .

4. *For Fig. 4c it would be better to choose a replicate with visible *parC*-beads, like in Fig. 4b, since yellow spots at the ends and sometimes the middle of the filaments are also detected in samples without beads (Suppl. Fig. 5 and 6). According to literature, the filaments stabilized by ParR and the DNA are longer because the complex acts as a cap and polymers attach to each other in an antiparallel manner. The protein in this manuscript seems to behave according to a different mechanism, therefore it is not possible to indirectly prove a bidirectional segregation event by polymer length. Also, labels which allow to distinguish between beads and lipids would be necessary. During image acquisition or processing, filaments were distorted (visible outside of GUVs, Fig. 3f).*

Thank the reviewer for the comments and valuable suggestion. We replaced the Figure 4c with replicate data with visible *parC*-beads in page 16 as below. We also replaced Figure 3f with the replicate in page 13 as below.

Page 16:

Figure 4 c Confocal time series images of two *parC*-beads segregated by ParM filament in a GUV that contained ParM (9.6 μM), ParR (53.6 μM), green *parC*-beads, 100 mM KCl, 30 mM Tris-HCl, 2 mM MgCl_2 , and 1 mM DTT. ATP (1 mM) was released from the outside upon laser irradiation (405 nm, 0.3

mW, 5 s). The scale bars are 10 μm .

Page 13:

Figure 3 f Schematic and confocal microscopy images of ParM filaments before (left column images) and after (right column images) laser irradiation (561 nm, 0.7 mW, 5 s). The scale bars are 10 μm .

5. *Coupling the ParMRC system to PURE is a good approach to ensure functional offspring of dividing cells. A fluorescent protein (eGFP) is successfully produced by PURE after dividing GUVs. However, true evidence for an active role of ParMRC in the segregation of the eGFP gene is missing. Dreher et al. (2021) also receive two fluorescent daughter cells upon Ce6-mediated division without the use of any segregation system. And since large filaments are formed even in the absence of ParR and parC, it is not clear if and how many plasmids are captured and segregated by the filament in individual GUVs. The authors could prove their system's ability to actively deliver DNA to offspring by performing a statistical analysis on the daughter cell's fluorescence signals: GUVs with the ParMRC system should have a significantly higher amount of daughter cells with equal fluorescence intensities compared to vesicles without the protein system. Generally, data confirming the reproducibility of the system should be provided. How many GUVs of a population are able to form a single filament of desired length, contain the right amount of DNA and can separate two parC-sequences?*

Thank the reviewer for the comments and valuable suggestions. We confirmed that all *parC* plasmids were attached on the magnetic beads, and no free plasmids were found in the solution (Supplementary Figure 28). The amount of *parC*-eGFP DNA on each bead was estimated to be ~ 0.87 ng, which was enough trigger eGFP expression using PURE systems inside daughter GUVs. We followed the suggestion and carried out control experiments for eGFP expression in daughter GUVs with no active segregation (ParMRC system). The corresponding descriptions were added as below in page 28 and page 19.

Page 28:

The GUVs were collected at the bottom of the tube and used for subsequent experiments. **ParMRC-containing GUVs were prepared through a similar method as that used for the ParM-containing**

GUVs. Since the amount of eGFP plasmid on each bead (~ 0.87 ng) is sufficient for expression in GUVs containing the PURE system, we encapsulated two *parC* beads in each GUV as much as possible; as a result, the beads bound to the two ends of ParM filaments, avoiding the free unbonded beads in the GUV.

Page 19:

In comparison, no green fluorescence was observed at 37 °C for 3 h inside daughter cells when the mother cells did not contain the PURE system (Supplementary Figure 24). Without the ParMRC system, the *parC*-eGFP DNA beads were randomly distributed into two daughter GUVs (Supplementary Figure 25), which resulted in a significant difference in the expression rate between the two daughter GUVs (Supplementary Figure 26). In contrast, the expression rates of each daughter GUV with the ParMRC system were almost identical (Supplementary Figure 26), which further confirmed the role of the ParMRC system in segregating genetic information in both daughter GUVs.

Supplementary Figure 25. Confocal microscopy images of GUV division and follow-on eGFP expression in two daughter GUVs without ParMRC system. Time-dependent fluorescence images of GUV division at 0 s (a), 43 s (b), 86 s (c), and eGFP expression at 37 °C in daughter cells at 90 min (d), 180 min (e), 210 min (f), respectively. The scale bars are 10 μm.

Supplementary Figure 26. The normalized fluorescence intensity of eGFP in daughter GUV 1 and GUV 2 with or without ParMRC system. The control experiments represent the condition of eGFP expression in daughter GUV 1 and GUV 2 without the ParMRC system. The normalized fluorescence intensities of eGFP are from three independent samples. Data are presented as the

mean values \pm SDs, n=3. Statistical analyses were carried out by unpaired two-tailed student's t-test. **p < 0.01. Source data are provided as a Source Data file. The ParMRC experiments represent the condition of eGFP expression in daughter GUV 1 and GUV 2 with the ParMRC system. The normalized fluorescence intensities of eGFP are from three independent samples. Data are presented as the mean values \pm SDs, n=3. Statistical analyses were carried out by unpaired two-tailed student's t-test. NS, not significant (P = 0.9670).

6. *In Fig. 5b it is not clear why the eGFP DNA is expected to be visible at the ends of the filament. A biotin-modification of the DNA is mentioned but it is not explained why it exists and what it is supposed to bind to. Generally, the method part of the manuscript lacks important information. The design of ParM is not clear, since it is not noted which terminal end contains the His-tag, if the tag is cleavable or not and if there is a second modification for the fluorescence label. Cleavage of the tag, protein labelling, and coverslip preparation are entirely left out. Percentages of SDS-PAGES are not indicated.*

Thank the reviewer for the comments and valuable suggestions. We followed the suggestion and added the missing contents in page 23, and page 24 as below.

Expression and Purification of ParM and ParR. ... The ParM gene was synthesized and cloned and inserted into the expression vector pET28a (a plasmid encoding an N-terminal 6-histidine tag) to generate the ParM plasmid (supplementary methods 1), which was subsequently transformed into *E. coli* BL21 (DE3) cells at a certain concentration (~10 ng). ... The purities of ParM and ParR were greater than 99%, as determined by densitometric analysis with ImageJ. The N-terminal 6-histidine tag was not cleaved before use. The extracted proteins were stored at -80 °C.

Labelling of ParM with fluorescent dyes. Five additional amino acids (GSKCK) were added to the C-terminus of the ParM gene for fluorescent labelling of the Cy3 maleimide dye. Cy3 maleimide dye (16 μ M) was mixed with ParM solution (96 μ M) for 10 minutes at 25 °C, followed by the addition of 10 mM DTT to quench the reaction. The free Cy3 maleimide dye molecules were removed by centrifugation in an ultrafiltration tube (10 kDa) at 12000 rpm for 20 minutes in a mixture of 260 mM sucrose, 30 mM Tris-HCl, 2 mM MgCl₂, 1 mM DTT, and 100 mM KCl at pH 7.5 three times. The samples were stored at -80 °C.

Coverslip preparation. The coverslips were cleaned by ultrasonication in dichloromethane for 15 min, dried under a stream of nitrogen, and rinsed in Milli-Q water before being immersed in piranha solution (70:30, v/v, H₂SO₄:H₂O₂) for 5 min. The coverslips were subsequently washed with Milli-Q water and ethanol. The coverslips were stored in ethanol until use. The polymerization of ParM and subsequent *parC*-beads segregation in solution were performed using these treated coverslips.

7 *Besides these severe technical shortcomings, the manuscript suffers from a very poor language quality. The immense number of grammar and spelling errors make it difficult to understand what Zhao and Han want to express. In the introduction it is repeatedly not clear if the literature is reviewed or something is claimed to be shown in the manuscript (e.g., l. 36-39). Some sentences are out of context and therefore not clear to the reader (l. 32-33).*

Moreover, important publications on ParMRC in the context of synthetic biology are not mentioned.

Thank the reviewer for the comments. The English has been improved by professional language editing service “Nature publishing group language editing service” to improve the clarity of the manuscript. We revised the introduction part as below in page 2, page 3 and page 4.

Cell division is a key feature of life¹, and it is crucial that daughter cells inherit accurate and correct genomic material to maintain the species². Eukaryotes possess complicated chromosome segregation machinery in mitotic spindles³⁻⁵. In prokaryotes, relatively simple mechanisms are used to partition newly replicated DNA⁶⁻⁹. The ParMRC partitioning system is among the most representative systems for bacterial DNA segregation^{6,10-12}. This system consists of three elements, namely, ParM (an actin-like protein), ParR (an adapter protein) and *parC* (a centromere-like site)¹³⁻¹⁵. ParM is an ATPase that can bind ParR, while ParR can specifically bind to *parC*. The genomic materials (plasmids) are segregated into opposite poles of cells by the polymerization of ParM with the help of ParR and *parC* in the presence of ATP¹⁶. **This process is important because it ensures that genetic material is evenly distributed among daughter cells prior to division. In a previous study, bipolar elongation of ParM filaments was observed in a bacterial cell through immunofluorescence microscopy¹⁷. Through time-lapse fluorescence microscopy, researchers observed dynamic segregation of plasmids towards the poles of cells by ParM polymerization, which was facilitated by ParR¹⁸. Once plasmids reach the poles of the cell, the ParM filaments disassemble spontaneously and distributes plasmids to each daughter cell after cell division. In addition to the in vivo study on the function of the ParMRC system, the *parC* beads were shown to move in the opposite direction following ParM polymerization with the help of ParR in vitro¹⁹. The ParMRC partitioning system was successfully reconstituted into water-in-oil droplets to realize in vitro DNA segregation²⁰, which provided the possibility for plasmid segregation and inheritance in artificial cells.**

Building artificial cells with true-to-life functionality is an ambitious goal in synthetic biology²¹⁻²⁵. The inheritance of genetic material into daughter cells after division is an essential step towards the construction of a minimal cell. **The division of artificial cells is triggered by mechanical force²⁶, osmotic pressure^{27,28}, pH^{29,30} and active molecules³¹. The vesicles are symmetrically split by the sharp edge of a wedge-shaped splitter inside a microfluidic channel, which results from mechanical force²⁶. This method provided a strategy for symmetric, quick, efficient vesicle division. Phase separation of lipids in the membrane of artificial cells provides a division plane for dividing mother vesicles into two daughter vesicles through osmotic pressure in a controllable manner²⁸. Oleic acid in the inner leaflet of the GUVs (POPC/oleic acid) bilayer was deprotonated by the enzymatic reaction of urea-urease inside the vesicle, which enabled inner leaflet area variation and consequently led to division with the help of osmotic pressure³⁰. All the abovementioned vesicle divisions did not involve the natural division protein machinery. Bacterial division proteins including MinC, MinD, MinE, FtsA, and FtsZ were reconstituted into lipid vesicles³²⁻³⁴, attempting to divide artificial cells. Currently, the division of artificial cells is mostly designed for shape splitting, and some divisions are accompanied by DNA redistribution^{27,31,35,36}. GUVs that contain DNA grow and divide into daughter vesicles by synthesizing and incorporating phospholipids in situ upon the addition of vesicular membrane precursors³¹. DNA inside the ‘nucleus’ of eukaryote-like artificial cells is redistributed into two daughter artificial cells after division via osmotic stress²⁷.**

To date, researchers have found that all DNA redistribution in daughter artificial cells is random and passive and does not involve protein machinery **or further translation of genetic information into DNA.**

8 *Generally, all references should be checked once again since their choice is not always clear. The authors reference articles which do not support their results without indicating this to the reader. Examples are already described for the purification and ATPase assay part. Apart from poorly discussing their results, they should make sure to quote original literature: As a reference for the complexity of eukaryotic mitotic spindles (l. 23-24) they reference a paper about actin-deformed artificial cells to mimic muscle contraction published by their own group (Li et al. 2022).*

Thank the reviewer for the comments and valuable suggestions. We have changed the reference for the purification from (Garner et al. 2007) to (Jiang et al. 2016, Koh et al. 2019), changed the references for the complexity of eukaryotic mitotic spindles from (Hurtgen et al. 2007, Li et al. 2022) to (Duro et al. 2015, Oliferenko et al. 2018, Heald et al. 2000), and deleted (Forde et al. 2017) due to the inconsistency of ParM origin. We also checked all references and make sure they were correctly cited.

9 *Most importantly, the general narrative of the manuscript needs to be fixed, as the authors do not really mimic bacterial cell division. The *GUV* is split via *Ce6*-mediated division, i.e., achieved by photoinduced oxidative stress and osmotic shock, not based on any biology. Reactive oxygen produced by this is harmful to cell components (like DNA, proteins and lipids). *Ce6*-mediated division has previously been used by Dreher et al. (2021) as a chemical strategy to divide DNA encapsulating vesicles. Thus, the bacterial segregation machinery *ParMRC* discussed here does not support the division of the vesicle but rather acts as an obstacle during *Ce6*-mediated division, therefore being eliminated by laser irradiation prior to division. Therefore, the term 'bacterial cell division' is misleading.*

Thank the reviewer for the comments and valuable suggestion. We fully agree with the reviewer and changed “bacterial division mimicry” to “bacterial plasmid segregation and inheritance mimicry” throughout the manuscript as below.

Page 1:

Investigation of artificial cells containing the Par system for bacterial plasmid segregation and inheritance mimicry

Page 1:

Using a PURE system, we translate eGFP DNA into enhanced green fluorescent proteins in daughter cells, and bacterial plasmid segregation and inheritance are successfully mimicked in artificial cells. Our results could lead to the construction of more sophisticated artificial cells that can reproduce with genetic information.

Page 4:

Herein, we mimicked bacterial plasmid segregation and inheritance by using the ParMRC system to segregate DNA at the two poles of artificial cells.

Page 5:

Figure 1. Schematic illustration of artificial cells containing the ParMRC system and PURE system for bacterial plasmid segregation and inheritance mimicry.

Page 17:

Bacterial plasmid segregation and inheritance mimicry using artificial cells containing the ParMRC system and PURE system

Page 18:

The full mimicry of bacterial plasmid segregation and inheritance was completed using artificial cells that contained the ParMRC system and PURE system (Figure 5c).

Reviewer #4 (Remarks to the Author):

We really appreciated the review's comments and valuable suggestion, which greatly improved the quality of our manuscript.

Reviewers' Comments:

Reviewer #1:

Remarks to the Author:

I have gone through the comments of Reviewer 2 (not me) and the author's responses. I understand, where the criticism of Reviewer 2 is coming from. The manuscript is essentially a combination of lots of different aspects of previous work (and I don't say this to devalue the work!), the previous work is cited. However, instead of acknowledging explicitly that "pore formation with Ce6 was done according to xxx (ref)" and "splitting of ParM was done according to...", they make it sound like they invented the mechanism. This gives not only a false sense of novelty and a lack of credit, but also leads to the critical questions of reviewer 2: If these things were indeed novel, much more characterization would be needed. So in the best interests of the authors, I encourage them to give appropriate credit by explicitly stating that the mechanisms were copied from the previous papers.

I still think (and Reviewer 2 seems to agree) that the combination of the different aspects warrants publication in Nature Communications. To my best knowledge, the comments have in large parts been addressed adequately. However, the previous work, upon which the paper builds, has to be cited more explicitly to not give the false impression that the authors came up with everything.

Reviewer #3:

Remarks to the Author:

The revised version of the manuscript by Zhao and Han demonstrates substantial improvements, as the authors have successfully addressed many open questions and greatly improved the writing.

However, a few weaknesses still persist:

The sentence 'After the ParM filament splits, the cells are divided into two daughter cells [...]' (l. 15) is misleading, as it still creates the false impression of filament splitting being an integral part of the segregation process, even though filaments are being split by the experimenter via laser irradiation. Since autonomous division of vesicles would be an important achievement on its own, it should be emphasized that the fission here is actually externally induced.

The authors have successfully improved their purification protocol of ParM yielding higher purity and protein functionality of the protein. However, the manuscript still contains results obtained with the deficient protein. As an example, Fig. 2e showing P_i release after 1 hour is the same as in the old version, even though the pure ParM has a much higher ATPase activity matching the literature value. Additionally, since the 'old' ParM contained lots of protein impurities, the concentrations used for the experiments could be significantly different. If this is the case, reproducing the results may lead to confusion, since the reader doesn't know about the deficient ParM used for these experiments when reading the revised manuscript.

Finally, the statistical power of the distribution is limited, since only three GUVs were tested in each condition. Given that cell division is controlled by the experimenter, caution must be exercised in the experimental setup and in interpreting the results. For the samples with ParMRC, vesicles which contained precisely two segregated beads were chosen, ensuring a 100% success rate under controlled division. Here, it remains unclear in how many vesicles within a population this event of successful segregation can be observed and whether filaments are formed in each vesicle. Despite the controlled division, it is surprising that successful segregation is very unfavourable in vesicles without ParMRC. Based on the supplemental figure it seems that the beads are quite diffusive, making segregation indeed difficult. Therefore, it would be better to split GUVs into equally sized vesicles to avoid diffusion into the much larger daughter. Hence, the advantage of ParMRC is not fully convincing.

Reviewer #4:

Remarks to the Author:

For the sake of clarity, the comments of the reviewer have been collated in black, and our response to each comment appears in blue. All the changes to the manuscript are highlighted in red.

Reviewer #1 (Remarks to the Author):

I have gone through the comments of Reviewer 2 (not me) and the author's responses. I understand, where the criticism of Reviewer 2 is coming from. The manuscript is essentially a combination of lots of different aspects of previous work (and I don't say this to devalue the work!), the previous work is cited. However, instead of acknowledging explicitly that "pore formation with Ce6 was done according to xxx (ref)" and "splitting of ParM was done according to...", they make it sound like they invented the mechanism. This gives not only a false sense of novelty and a lack of credit, but also leads to the critical questions of reviewer 2: If these things were indeed novel, much more characterization would be needed. So in the best interests of the authors, I encourage them to give appropriate credit by explicitly stating that the mechanisms were copied from the previous papers.

I still think (and Reviewer 2 seems to agree) that the combination of the different aspects warrants publication in Nature Communications. To my best knowledge, the comments have in large parts been addressed adequately. However, the previous work, upon which the paper builds, has to be cited more explicitly to not give the false impression that the authors came up with everything.

Thank the reviewer for the comments and valuable suggestion. We added below contents in page 10 and page 12 to eliminate the false impression.

Page 10:

To trigger the polymerization of ParM inside GUVs, ATP molecules were introduced into GUV from the outside upon light irradiation in the presence of Chlorin e6 (Ce6) according to the protocol described in previous paper³⁵ (Supplementary Figure 1).

Page 12:

Here, the ParM filaments were precisely cut into two parts both in solution (Supplementary Figure 15) and inside the GUV (Figure 3f) after strong laser irradiation (561 nm, 0.7 mW, 5 s) according to the method described in previous work¹⁹, due to the laser-induced photochemical reaction of Cy3 dyes which were attached to ParM via chemical bonds.

Reviewer #3 (Remarks to the Author):

The revised version of the manuscript by Zhao and Han demonstrates substantial improvements, as the authors have successfully addressed many open questions and greatly improved the writing.

However, a few weaknesses still persist:

- (1) The sentence ‘After the ParM filament splits, the cells are divided into two daughter cells [...]’ (l. 15) is misleading, as it still creates the false impression of filament splitting being an integral part of the segregation process, even though filaments are being split by the experimenter via laser irradiation. Since autonomous division of vesicles would be an important achievement on its own, it should be emphasized that the fission here is actually externally induced.***

Thank the reviewer for the comments and valuable suggestion. We added below contents in page 1 to eliminate the false impression.

After the ParM filament splits, the cells are **externally induced** to divide into two daughter cells that contain *parC*-eGFP DNA **by osmotic pressure and laser irradiation**.

- (2) The authors have successfully improved their purification protocol of ParM yielding higher purity and protein functionality of the protein. However, the manuscript still contains results obtained with the deficient protein. As an example, Fig. 2e showing P_i release after 1 hour is the same as in the old version, even though the pure ParM has a much higher ATPase activity matching the literature value. Additionally, since the ‘old’ ParM contained lots of protein impurities, the concentrations used for the experiments could be significantly different. If this is the case, reproducing the results may lead to confusion, since the reader doesn’t know about the deficient ParM used for these experiments when reading the revised manuscript.***

Thank the reviewer for the comments and valuable suggestion. We followed the suggestion and carried out the experiments with high purity proteins and replaced relevant figures including Supplementary Figure 3, Figure 2e-f, Supplementary Figure 4, Figure 3b, Supplementary Figure 6, Figure 3c-e, and Figure 4b. Below contents were added in page 6, page 7, page 9, page 10, page 11 and page 12.

Page 6:

Pi release linearly increased with time in the first phase and then stabilized after 4 mM ATP was hydrolysed by different concentrations of ParM (Supplementary Figure 3).

Supplementary Figure 3. Phosphate (Pi) release of ATP catalyzed by ParM as a function of time. The phosphate release of ATP (4 mM) at 270 min was 2.7 µM, 12.3 µM, 26.3 µM, 43.5 µM, 61.1 µM, 66.2 µM, and 73.9 µM catalyzed by ParM with the concentrations of 0 µM, 0.46 µM, 0.92 µM, 1.84 µM, 2.8 µM, 3.7 µM, 4.6 µM, respectively.

Page 7:

ParR addition increased the ATPase activity of ParM (Figure 2f) by ~1.47, 1.61, 1.62, 1.65, and 1.69-fold with ParR concentrations of 7.7 µM, 23.0 µM, 38.4 µM, 76.8 µM, and 115.1 µM, respectively.

e

f

Figure 2e Phosphate release of ATP (4 mM) catalysed by different ParM concentrations (0 µM, 0.46 µM, 0.92 µM, 1.84 µM, 2.8 µM, 3.7 µM, 4.6 µM) within 1 hour. The phosphate release was obtained from three independent samples. The data are presented as the mean values ± SDs; n=3. **f** The ATPase activity of ParM (4.6 µM) enhanced by ParR (7.7 µM, 23.0 µM, 38.4 µM, 76.8 µM, 115.1 µM); the pink, black, and blue columns represent the ATPase activity of ParM, ParR, and ParM-ParR, respectively. The ATPase activities of ParM, ParR, and ParM-ParR were obtained from three independent samples. The data are presented as the mean values ± SDs (n = 3).

Page 9:

ParM dots were generated with ATP concentrations less than 0.2 mM at a ParM concentration of 19.1 µM (Supplementary Figure 4).

Supplementary Figure 4. Influence of ATP concentration on ParM polymerization. Fluorescence microscopy images of ParM (19.1 μM) polymerization at the ATP concentrations of 0 mM (a), 0.15 mM (b), 0.20 mM (c), 0.33 mM (d), 0.40 mM (e), and 1 mM (f) within 5 min. The polymerized buffer contains 30 mM Tris-HCl, 2 mM MgCl_2 , 1 mM DTT, 100 mM KCl at pH 7.5. The scale bars are 20 μm .

Page 10:

The average lengths of the filaments were $5.0 \pm 2.2 \mu\text{m}$, $13.9 \pm 5.8 \mu\text{m}$, $17.6 \pm 8.3 \mu\text{m}$, $35.4 \pm 16.7 \mu\text{m}$ and $46.1 \pm 24.2 \mu\text{m}$ at ParM concentrations of 2.4 μM , 4.8 μM , 9.6 μM , 14.3 μM and 19.1 μM , respectively (Figure 3b, Supplementary Figure 6).

Figure 3b ParM filament length at ParM concentrations of 2.4 μM , 4.8 μM , 9.6 μM , 14.3 μM , and 19.1 μM . The concentration of ATP was 1 mM. The lengths of ParM were determined from 100 independent samples at each concentration. $n = 100$. The central line corresponds to the median. The black dots correspond to the mean values. The lower and upper hinges of the boxes correspond to the 25th and 75th percentiles, respectively, and the whiskers represent the $1.5 \times$ interquartile range extending from the hinges. Statistical analyses were carried out by an unpaired two-tailed Student's t test. $**P < 0.001$, $****P < 0.000001$.

Supplementary Figure 6. Fluorescence microscopy images of ParM polymerization at different concentrations of ParM. ParM (0.48 μM , 2.4 μM , 4.8 μM , 9.6 μM , 14.3 μM , 19.1 μM) was polymerized in buffer (30 mM Tris-HCl, 2 mM MgCl_2 , 1 mM DTT, 100 mM KCl, pH 7.5) triggered with 1 mM ATP. The scale bars are 20 μm .

Page 11:

The growth of ParM filaments was observed inside lipid-protected droplets from short filaments ($5.9 \pm 2.5 \mu\text{m}$) at 0 s to long filaments ($23.5 \pm 3.3 \mu\text{m}$) at 172 s (Figure 3c, video S1). The length of the ParM filament was plotted as a function of time (blue, Figure 3e), from which an average growth rate of $\sim 41 \pm 7$ monomers s^{-1} was obtained assuming a monomer length of 2.45 nm¹⁹.

Page 12:

From the growth curve (red curve in Figure 3e), the elongation rate of the ParM filament was $\sim 27 \pm 4$ monomers s^{-1} .

Figure 3c Confocal microscopy images of a ParM filament as a function of time inside the lipid-protected droplet (green dotted circle for eye-guide) with a ParM concentration of 9.6 μM and an ATP concentration of 1 mM. The scale bars are 20 μm . **d** Confocal microscopy images of ParM filaments as a function of time inside the GUV with a ParM concentration of 9.6 μM . ATP (1 mM) outside the GUV membrane flew inside upon 5 s of laser irradiation (405 nm). Scale bars are 10 μm .

Figure 3e The length of ParM filaments inside lipid-protected droplets (blue curve) and GUV (red curve) as a function of time. The length of ParM filaments inside the lipid-protected droplet (blue curve) and GUV (red curve) were obtained from three independent samples. The data are presented as the mean values \pm SDs; $n = 3$.

Page 15:

The green dots at the two ends of the filament were observed to separate gradually to reach $7.4 \mu\text{m}$ at 172 s in solution (Figure 4b), which confirmed that the DNAs were segregated by ParM filament growth.

Figure 4b Confocal time series images of the segregation of two *parC*-beads (green) through ParM (red fluorescence) polymerization in solution. The white arrows indicate the *parC*-beads. The solution contained ParM ($9.6 \mu\text{M}$), ParR ($53.6 \mu\text{M}$), green *parC*-beads, 1 mM ATP, 100 mM KCl, 30 mM Tris-HCl, 2 mM MgCl_2 , and 1 mM DTT. The scale bars are $5 \mu\text{m}$.

(3) *Finally, the statistical power of the distribution is limited, since only three GUVs were tested in each condition. Given that cell division is controlled by the experimenter, caution must be exercised in the experimental setup and in interpreting the results. For the samples with ParMRC, vesicles which contained precisely two segregated beads were chosen, ensuring a 100% success rate under controlled division. Here, it remains unclear in how many vesicles within a population this event of successful segregation can be observed and whether filaments are formed in each vesicle. Despite the controlled division, it is surprising that successful segregation is very unfavourable in vesicles without ParMRC. Based on the supplemental figure it seems that the beads are quite diffusive, making segregation indeed difficult. Therefore, it would be better to split GUVs into equally sized vesicles to avoid diffusion into the much larger daughter. Hence, the advantage of ParMRC is not fully convincing.*

Thank the reviewer for the comments and valuable suggestion. We found the filament formation

rate is about 100% in the vesicles with ParMRC system. Among the 11 vesicles with ParM filament formation, 9 vesicles were found with the bead segregation by the ParM filaments. The beads segregation rate with ParMRC system is about 82%. Due to the small size of bead inside a large volume vesicle, the distribution of beads is random, which causes the chance of one bead in each daughter vesicle without ParMRC system is very low after division. We found only 1 vesicle were divided into daughter vesicles with one bead in each daughter vesicle out of 11 vesicles that divided into equally sized daughter vesicles. The bead segregation rate is ~9% without ParMRC system. The successful bead segregation rate is much higher in the vesicles containing ParMRC system than those containing no ParMRC system, which proved the importance of ParMRC system for beads segregation.

We replaced the supplementary figure 25 and 26. Below contents were added in page 19.

Page 19:

Without the ParMRC system, the *parC*-eGFP DNA beads were randomly distributed into two daughter GUVs (Supplementary Figure 25), which resulted in a significant difference in the expression rate between the two daughter GUVs (Supplementary Figure 26). In contrast, the expression rates of each daughter GUV with the ParMRC system were almost identical (Supplementary Figure 26). **Without the ParMRC system, only 1 GUV was divided into daughter GUVs with one *parC*-eGFP DNA bead in each daughter GUV out of 11 GUVs that divided into equally sized daughter GUVs. The bead segregation rate is ~9% without ParMRC system. With ParMRC system, the bead segregation rate is ~82% (n=11). The successful bead segregation rate is much higher in the GUVs containing ParMRC system than those containing no ParMRC system, which further confirmed the role of the ParMRC system in segregating genetic information in both daughter GUVs.**

Supplementary Figure 25. Confocal microscopy images of GUV division and follow-on eGFP expression in two daughter GUVs without ParMRC system. Time-dependent fluorescence images of GUV division at 0 s (a), 43 s (b), 86 s (c), and eGFP expression at 37 °C in daughter cells at 60 min (d), 180 min (e), 240 min (f), respectively. The scale bars are 10 μm.

Supplementary Figure 26. The normalized fluorescence intensity of eGFP in daughter GUV 1 and GUV 2 with or without ParMRC system. The control experiments represent the condition of eGFP expression in daughter GUV 1 and GUV 2 without the ParMRC system. The normalized fluorescence intensities of eGFP are from three independent samples. Data are presented as the mean values \pm SDs, $n=3$. Statistical analyses were carried out by unpaired two-tailed student's t-test. $***p < 0.01$. Source data are provided as a Source Data file. The ParMRC experiments represent the condition of eGFP expression in daughter GUV 1 and GUV 2 with the ParMRC system. The normalized fluorescence intensities of eGFP are from three independent samples. Data are presented as the mean values \pm SDs, $n=3$. Statistical analyses were carried out by unpaired two-tailed student's t-test. NS, not significant ($P = 0.9670$).

Reviewer #4 (Remarks to the Author):

We really appreciated the review's comments and valuable suggestion, which greatly improved the quality of our manuscript.

Reviewers' Comments:

Reviewer #3:

Remarks to the Author:

The authors have performed experiments with the newly purified protein showing a drastically higher ATPase activity. This is, on one hand, very convincing. On the other hand, they should ideally now check whether this affects the results quantitatively (or even qualitatively), and if so, repeat the most important experiments with the new protein batch. E.g., Fig. 3a still shows ParM concentrations that potentially differ from concentrations with the pure protein.

Is the strong deviation from a 1:1 distribution of beads into daughter cells without segregation machinery expected? Maybe the authors can discuss this briefly, or ideally explain the observation based on relevant literature.

For the sake of clarity, the comments of the reviewer have been collated in black, and our response to each comment appears in blue. All the changes to the manuscript are highlighted in red.

Reviewer #3 (Remarks to the Author):

- The authors have performed experiments with the newly purified protein showing a drastically higher ATPase activity. This is, on one hand, very convincing. On the other hand, they should ideally now check whether this affects the results quantitatively (or even qualitatively), and if so, repeat the most important experiments with the new protein batch. E.g., Fig. 3a still shows ParM concentrations that potentially differ from concentrations with the pure protein.*

Thank the reviewer for the comments and valuable suggestion. We followed the suggestion and carried out the experiments of Figure 3a and Supplementary Figure 19 with high purity proteins. The corresponding figures were updated as below.

Page 13:

The threshold ATP concentration for the formation of ParM filaments varied with ParM concentration (Figure 3a), in which the green dots indicate the successful formation of ParM filaments under these conditions and the cross symbols indicate the failure of ParM filament formation.

Figure 3a Phase diagram of ParM filament formation using ATP concentration (0.15 mM, 0.2 mM, 0.33 mM, 0.4 mM, 1 mM) and ParM concentration (0.6 µM, 1.5 µM, 2.2 µM, 3.0 µM, 5.9 µM, 11.7 µM) as parameters.

Page 17:

After the *parC*-beads were segregated at two poles of GUV by the splitting of the ParM filament, the artificial cells were divided into two daughter GUVs that contained genetic materials (*parC*-beads) (Supplementary Figure 19).

Supplementary Figure 19. The confocal time series images of *parC*-beads redistribution through GUV division. The *parC*-beads positioned to two poles of GUV by the split of ParM filament, and redistributed into two daughter GUVs after division. GUV membrane was labeled with NBD-PE (green fluorescence). ParM filament labeled with cy3 (red fluorescence). *parC*-beads labeled with SYBR Green I (green fluorescence). The white arrows referred to the *parC*-beads. The scale bars are 10 μm .

2. Is the strong deviation from a 1:1 distribution of beads into daughter cells without segregation machinery expected? Maybe the authors can discuss this briefly, or ideally explain the observation based on relevant literature.

Thank the reviewer for the comments and valuable suggestion. In principle, the probability of 1:1 distribution of magnetic beads into daughter GUVs without segregation machinery is 50% in fully random condition. The bead segregation rate is ~82% with ParMRC system, which already proves the superiority of ParMRC system for plasmids segregation.

In this paper, the probability of one bead in each daughter GUV is 9% without ParMRC system. The asymmetric division also exists in real cells without active segregation systems. Mitochondria in eukaryotic cells were equally distributed to daughter cells by actin cables at cytokinesis; however, asymmetrical allocation of mitochondrial mass between daughter cells happened with the absence of actin cables^{1,2}. The high probability of two beads in one daughter GUV without ParMRC system may be caused by the gravity effect and weak magnetic interactions of magnetic beads. The density of magnetic beads is larger than that of aqueous solution, which enables more chance for two beads to locate in the lower half spherical GUV (Figure R1), thus increasing the probability of two beads in one daughter GUV after division. The second reason may be due to the weak interaction between these two magnetic beads. During the process of preparing *parC*-beads, the removal of excess DNA was achieved by magnetic separation rack. This process may cause the beads to possess remaining weak magnetism, which cause two beads tend to close to each other, consequently increasing the probability of two beads in one daughter GUV. We observed the two beads gradually closing inside a GUV as a function of time (Figure R2).

We added below contents in page 19.

Figure R1. Schematic illustration of two beads sinking in a dumbbell shaped vesicle due to the gravity.

Figure R2. Time-dependent confocal microscopy images fluorescence images of the GUV containing the two *parC*-beads at 0 min, 5 min, 10 min, 15 min, respectively. The scale bars are 5 μm .

Page 19:

Without the ParMRC system, only 1 GUV was divided into daughter GUVs with one *parC*-eGFP DNA bead in each daughter GUV out of 11 GUVs that divided into equally sized daughter GUVs. The bead segregation rate is $\sim 9\%$ without ParMRC system. **This may be caused by gravity and weak magnetic interaction of two magnetic *parC*-eGFP DNA beads. The asymmetric division also exists in real cells without active segregation systems. Mitochondria in eukaryotic cells were equally distributed to daughter cells by actin cables at cytokinesis; however, asymmetrical allocation of mitochondrial mass between daughter cells happened with the absence of actin cables^{46,47}.** With ParMRC system, the bead segregation rate is $\sim 82\%$ ($n=11$). The successful bead segregation rate is much higher in the GUVs containing ParMRC system than those containing no ParMRC system, which further confirmed the role of the ParMRC system in segregating genetic information in both daughter GUVs.

Reference

- 1 Moore, A. S. et al. Actin cables and comet tails organize mitochondrial networks in mitosis. *Nature* 591, 659+ (2021).
- 2 Rohn, J. L. et al. Myo19 Ensures Symmetric Partitioning of Mitochondria and Coupling of Mitochondrial Segregation to Cell Division. *Current Biology* 24, 2598-2605 (2014).

Reviewers' Comments:

Reviewer #3:

Remarks to the Author:

The authors have addressed all concerns, and the study can be published as is.